# Scene Graph Thinking: Reinforcing Structured Visual Reasoning for Multimodal Large Language Models

Zhiwei Yang [*1] Yuanchen Wu [*1] Nan Zhang [1] Yucong Meng [2] Ke Yan [†1] Shouhong Ding [1]

## Abstract

Multimodal Large Language Models (MLLMs) have demonstrated strong perception and reasoning capabilities. However, most existing models focus on isolated objects and neglect structured relationships for efficient target navigation, limiting their performance on visually intensive tasks. To address this challenge, we introduce *Scene Graph Thinking (SaGe)*, a novel paradigm that enables fine-grained and structured visual reasoning through explicit scene-graph representations. Specifically, we first introduce an automated data engine that converts flat image–text corpora into structured scene graphs, where hierarchical entities constitute the nodes and diverse visual relations define the edges. Building upon this, we construct 120K high-quality training data by sampling reasoning traces from scene graphs. Then two-stage graph-aligned post-training paradigms are introduced, where supervised fine-tuning internalizes MLLMs with structured reasoning, and subsequent reinforcement fine-tuning proposes node-as-proxy graph rewards to consolidate efficient graph exploration. With curated data and graph-aligned training, our approach achieves significant improvements across eight multimodal benchmarks, demonstrating strong effectiveness on fine-grained perception and reasoning tasks.

## 1. Introduction

Multimodal Large Language Models (MLLMs) (Hurst et al., 2024; Jaech et al., 2024; Comanici et al., 2025) have advanced rapidly in recent years, showing growing competence in multimodal perception (Lai et al., 2024; Liu et al., 2025) understanding (Wang et al., 2023; Zhu et al., 2025b),

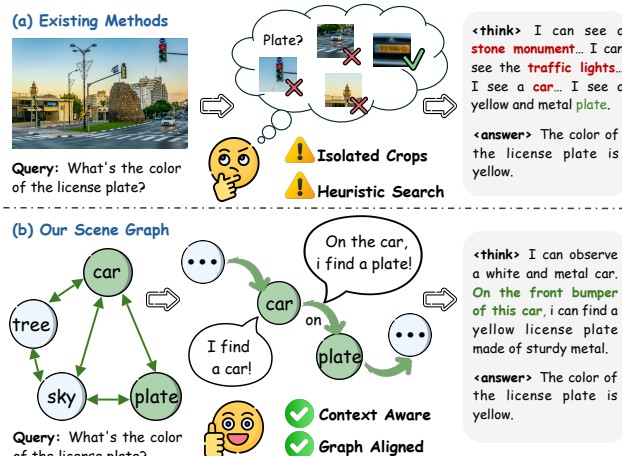

*Figure 1.* **Our motivation.** (a) Previous methods overlook structured relationships within the scene, leading to inefficient target navigation and suboptimal performance. (b) The proposed SaGe internalizes structured scene graphs to enable fine-grained visual reasoning, achieving more efficient and reliable performance.

and reasoning (Wang et al., 2025a; Shen et al., 2025). As instruction-following abilities and cross-modal transfer continue to improve, MLLMs are increasingly capable of handling a broad spectrum of tasks, driving their adoption in both general-purpose and specialized applications (Yin et al., 2024; Wang et al., 2025d). However, despite these advances, current MLLMs still struggle in complex visual scenarios and exhibit suboptimal performance in fine-grained analysis. (Zhang et al., 2025a; Wang et al., 2025b).

Recent milestones, such as OpenAI's o3 (OpenAI, 2025), pioneer to address above limitations by integrating visual information during the reasoning (Shao et al., 2024). Following these works, several studies (Zheng et al., 2025; Fan et al., 2025; Wang et al., 2025b) have adopted a *"think-with-image"* paradigm. This literature typically applies cropping and zoom-in operations to magnify target subjects, thereby enhancing fine-grained perception. While effective for local awareness, these methods **primarily focus on *isolated* entities and overlook *structured relationships* among scene elements** (Yang et al., 2025b). As shown in Figure 1 (a), this lack of relational and structural context leads to inefficient navigation to targets and suboptimal performance on holistic understanding, leaving fine-grained perception and

---

[*]Equal contribution . [†]Corresponding author. [1]Tencent Youtu Lab, Shanghai, China [2]Fudan University, Shanghai, China. Correspondence to: Ke Yan <kerwinyan@tencent.com>.

*Proceedings of the 43rd International Conference on Machine Learning*, Seoul, South Korea. PMLR 306, 2026. Copyright 2026 by the author(s).

reasoning still insufficiently explored.

To bridge this gap, we introduce **Scene Graph Thinking (SaGe)**, a paradigm that equips MLLMs with fine-grained and structured visual reasoning through explicit scene-graph representations (Xu et al., 2017; Min et al., 2025). As shown in Figure 1 (b), instead of relying isolated crops with heuristic searching, SaGe pre-organizes objects, their components, attributes, and relationships into a coherent scene graph. This representation serves as a structured data prior for generating reasoning trajectories. It enables the model to *perceive objects in context, reason over relational dependencies, and traverse structured pathways* through the scene, leading to more efficient and reliable visual reasoning.

Specifically, we first introduce **an automated data engine converting images with flat texts into structured scene graphs**, where objects in the image serve as nodes and visual relations define the edges. To enhance graph granularity, we design a sub-entity mining strategy that constructs hierarchical nodes and captures fine-grained object compositions. Each node is enriched with detailed attribute descriptions, precise spatial coordinates, and estimated depth ranges, forming an attribute-rich and 3D spatially-aware representation. Correspondingly, each edge explicitly models diverse relationships to connect related nodes with spatial, interaction, and semantic relations, resulting in a richly structured and semantically comprehensive scene graph.

To cultivate scene-graph–aligned reasoning, we construct **large-scale training data via LLM-guided graph sampling and traversal**. Sampling extracts both nodes and edges from graphs to form query-answer pairs: *node-centric queries* capture fine-grained attributes (e.g., color, material, state, location), while *edge-centric queries* emphasize relational reasoning (e.g., spatial layouts, interactions, ownership). Each query is augmented with a node-articulated chain-of-thought (CoT) (Wei et al., 2022), where nodes are articulated with entity names, bounding boxes, and depth cues as explicit reasoning evidence. Furthermore, graph traversal follows relational edges to synthesize multi-hop navigation traces and comprehensive scene captions, fostering target-oriented navigation. This process yields 120K structured samples for cold-start training, internalizing the model with robust graph-aligned reasoning abilities.

Building on this foundation, we further introduce a GRPO-based (Guo et al., 2025) reinforcement post-training stage to consolidate the structured reasoning. Two **complementary node-as-proxy graph rewards** are designed. A *node-relevance reward* guides the model to traverse semantically meaningful nodes aligned with the query intent, penalizing ambiguous or irrelevant graph traces. In parallel, a *node-grounded reward* is designed to promote visual attention by rewarding accurate spatial localization of the referenced entities. Together, these rewards guide the model to acquire

reasoning traces that are both semantically coherent and visually grounded, strengthening its ability to follow the scene graph during the reasoning. The main contributions of our work are listed as follows:

- **Scene Graph Thinking**: We introduce Scene Graph Thinking (SaGe), a paradigm that equips MLLMs with fine-grained and structured visual reasoning via explicit scene-graph representations.

- **Automated Data Engine**: We develop a pipeline that converts flat image–text corpora into structured scene graphs, producing 120K training data with attribute-rich, spatial-aware, and relation-guided CoT.

- **Two-Stage Post-Training**: We propose two graph-aligned training stages with supervised and node-based reinforcement finetuning, seamlessly internalizing the hierarchical graph reasoning into current MLLMs.

- **Strong Performance**: Extensive experiments demonstrate that our method achieves consistent gains across eight visually intensive benchmarks, with strong generalization and reasoning capabilities.

## 2. Related Works

### 2.1. Multi-modal Large Language Models

With the advancement of large language models (LLMs), integrating visual encoders via multimodal alignment pretraining (Radford et al., 2021; Cho et al., 2021; Wang et al., 2024) and instruction fine-tuning (Dai et al., 2023) has enabled Multimodal Large Language Models (MLLMs) to achieve strong task-level generalization. Early efforts primarily combined pretrained visual encoders with LLMs through lightweight projection layers or adapters (Li et al., 2023; Liu et al., 2023), enabling basic multimodal understanding. More recent models, such as Qwen2.5-VL (Bai et al., 2025b), LLaVA-OneVision (Li et al., 2024), and InternVL3 (Zhu et al., 2025a), further scale training by incorporating more diverse visual data, leading to substantial performance gains on vision–language benchmarks. However, despite this progress, vision–language models still struggle with fine-grained visual perception.

### 2.2. Fine-grained Visual Perception

Current MLLMs still struggle with fine-grained perceptual capabilities (Yang et al., 2025c; 2026; Wang et al., 2025c). To mitigate this, approaches such as o3 (OpenAI, 2025) incorporate visual information as a dynamic component of the reasoning process. Building on this idea, subsequent studies commonly adopt Group Relative Policy Optimization (GRPO) to incentivize zoom-in operations, thereby amplifying attention on target regions. (Zhang et al., 2025b; Hong et al., 2025; Wang et al., 2025b). Although effective for local details, these methods rely on commonsense-driven

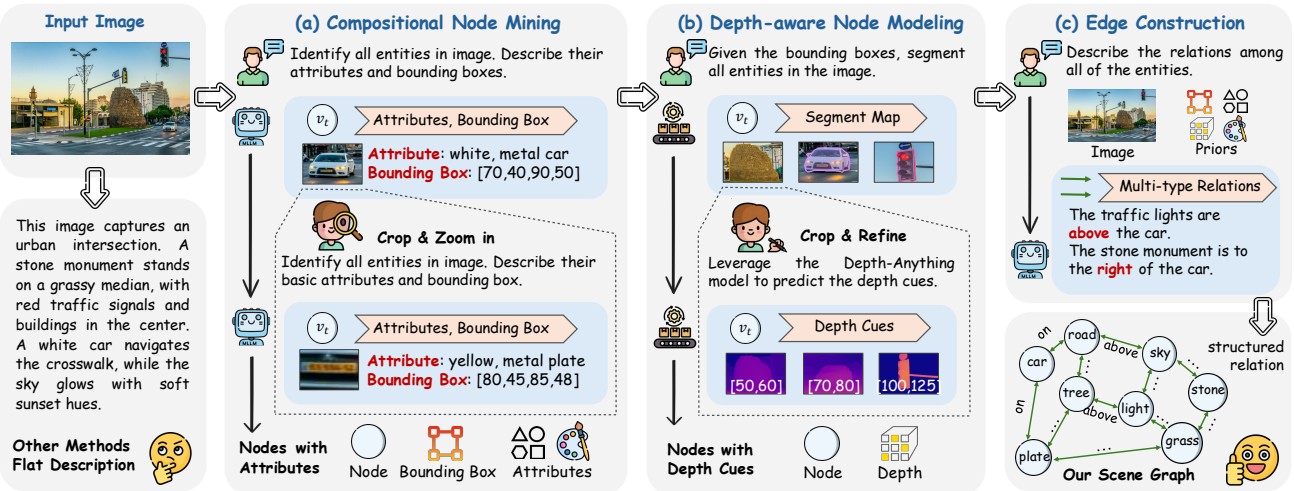

Figure 2. **Scene Graph Construction.** We convert each flat image–text pair into a *hierarchical* scene graph. Given an image $I_i$, (a) we first perform Compositional Node Mining, modeling all objects and their sub-components as nodes, each equipped with bounding boxes and attribute annotations. (b) We then build Depth-aware Node Modeling by assigning depth cues to each node. (c) Finally, edges are constructed based on node priors, connecting nodes with diverse semantic relationships. Details are provided in subsection 3.2.

search (Wu & Xie, 2024) and overlook structured dependencies among scene elements (Su et al., 2025; Li et al., 2025a; Fan et al., 2025). As a result, they incur redundant multi-turn navigation and struggle with holistic understanding. In contrast, we pre-construct a comprehensive scene graph that explicitly encodes objects, attributes, and multi-type relationships. By **reasoning over relational dependencies rather than heuristic exploration**, our approach facilitates more efficient and structured visual reasoning.

## 3. Methodology

### 3.1. Motivation

Most existing MLLMs rely on flat visual representations, which ignore the complex spatial and semantic relations of a scene. A traditional image-text dataset $\mathcal{D}_{\text{flat}}$ is denoted as:

$$\mathcal{D}_{\text{flat}} = \{(I_i, \mathcal{T}_i)\}_{i=1}^N, \qquad (1)$$

where $N$ is the sample number, $\mathcal{I}_i$ is the $i$-th image, and $\mathcal{T}_i$ is the textual annotation (e.g., question, or instruction).

To address this limitation, we propose *Scene Graph Thinking*, a paradigm that formalizes the flat image-text pair as a hierarchical scene graph $G_i = (V_i, E_i)$, where entities within $I_i$ are represented as nodes $V_i$ and their relationships are explicitly modeled as edges $E_i$. The structured texts $\mathcal{T}_i$ is then derived by sampling semantic subgraphs from $G_i$, ensuring the reasoning traces are grounded in the scene's hierarchy. The resulting dataset is formulated as:

$$\mathcal{D}_{\text{SG}} = \{(I_i, \mathcal{T}_i \mid G_i)\}_{i=1}^N, \qquad (2)$$

where $\mathcal{T}_i \mid G_i$ denotes that the texts are sampled from the topological structure of $G_i$. This formulation ensures that $\mathcal{T}_i$ encapsulates the connectivity of the scene, thereby serving

as the structured training data to endow the MLLMs with fine-grained and relation-aware reasoning.

### 3.2. Construction of Scene Graph

Our scene graph construction emphasizes fine-grained, spatially grounded, and relation-aware hierarchies, modeling subjects within the image as nodes and diverse relationships as edges. However, when applying existing MLLMs for graph construction, their **inherent limitations lead to three key challenges**: (1) limited fine-grained and compositional perception; (2) insufficient spatial understanding, particularly depth awareness; and (3) unreliable modeling of inter-entity relationships. To overcome these challenges, we develop a fully automated hybrid annotation pipeline.

**Compositional Node Mining.** As shown in Figure 2 (a), we first prompt Qwen2.5-VL-72B to detect salient objects together with their bounding boxes (bbox) and attributes in $I_i$, modeling them as a set of primary nodes $V_i$ in the scene graph. **To address *Challenge 1***, we introduce a compositional mining strategy that explores the internal structure of each primary node. Specifically, we crop and enlarge the visual region defined by each bbox. The zoomed region is then re-fed into the MLLM to identify sub-components together with their bbox and attributes. This procedure constructs node representations at multiple semantic scales, substantially enhancing the granularity of the scene graph.

**Depth-aware Node Modeling.** As shown in Figure 2 (b), we augment each entity with explicit depth cues **to address *Challenge 2***. We first generate a depth map for each image with Depth Anything (Yang et al., 2024). Since bbox cropping may introduce background noise for depth perception, we use it to prompt SAM for precise segmentation. After

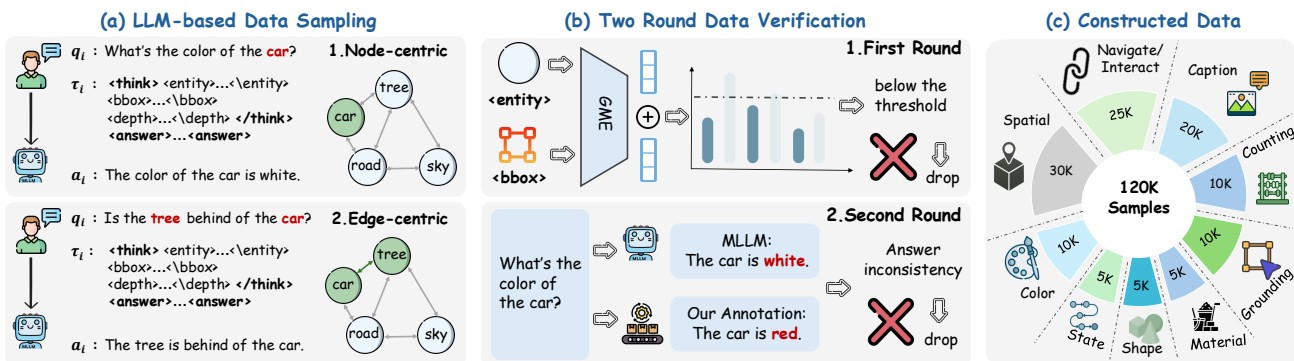

*Figure 3.* **Training Data Sampling and Verification.** We adopt (a) node- and edge-centric query sampling with a (b) two-round data verification strategy, (c) yielding 120K CoT-augmented training samples.

boundary erosion, we extract depth values within the mask to estimate a depth range for each entity, providing each node with reliable depth cues for 3D-aware reasoning. With the extracted attributes $a_t$, bbox $b_t$, and depth cues $d_t$, the $t$-th graph node $v_t$ is defined as:

$$v_t = (c_t,\ a_t,\ b_t,\ d_t), \tag{3}$$

where $c_t$ is the semantic category of the node, $b_t$ provides 2D coordinates of the bounding box $[x_t^{\min}, y_t^{\min}, x_t^{\max}, y_t^{\max}]$, and $d_t = [z_t^{\min}, z_t^{\max}]$ is the estimated depth range.

**Edge Construction via Node Priors.** As shown in Figure 2 (c), we build graph edges with the hieratically mined node-level priors (e.g., bounding boxes, depth cues, and component structure) rather than direct MLLM-based relation inference **to address *Challenge 3*.** These priors enable the construction of spatial, interaction, and semantic relations, explicitly linking relevant nodes and providing structured pathways for graph-based reasoning.

Through the above automated data engine, we generate structured scene graphs with fine-grained entities, depth-aware spatial grounding, and explicit relational structure.

### 3.3. Data Sampling and Verification

**LLM-based Data Sampling.** Based on the scene graphs, we automatically generate large-scale training data via LLM-guided graph sampling and traversal to foster hierarchical reasoning. To provide global context, we first use (Lu et al., 2025) to generate a holistic scene caption $C_i$ for each image as a global prior. Conditioned on $C_i$ and the scene graph $G_i$, GPT-OSS-120B $f(\cdot)$ (Agarwal et al., 2025) samples nodes and edges from the graph, as illustrated in Figure 3 (a).

For ***node-centric*** sampling, we generate attribute-focused queries (e.g., color, shape, and state), while ***edge-centric*** sampling emphasizes relational reasoning among connected nodes, such as spatial layouts and interactions. To strengthen graph-aligned reasoning, each query is paired with a node-articulated chain-of-thought (CoT) enclosed

in `<think>`. The referenced nodes are explicitly traced by entity names, bbox, and depth cues using `<entity>`, `<bbox>`, and `<depth>`, respectively. Beyond single-node queries, we further synthesize multi-hop navigation traces that traverse relational traces toward target nodes, as well as comprehensive scene captions connecting all nodes. Formally, the data sampling from graph is defined as:

$$(q_i, a_i, \tau_i) = f(G_i, C_i), \tag{4}$$

where $q_i$ is the question, $a_i$ is the answer or the caption, and $\tau_i = (v_1, v_2, \ldots, v_T)$ refers to reasoning trace articulated by $T$ visited nodes. These diverse reasoning patterns jointly promote fine-grained perception, effective relational navigation and structured graph-aligned reasoning.

**Two-round Data Verification.** To ensure the data quality, we adopt a *two-round* data verification, as shown in Figure 3 (b). First, we verify the consistency between each node's visual region and its semantic description using vision–text similarity. For each entity, we crop the bbox region and extract visual and textual embeddings with (Zhang et al., 2024). We discard samples containing nodes with low similarity scores, ensuring the accurate visual–semantic alignment. Second, we **validate reasoning integrity** by cross-verifying each query with a proprietary MLLM. It filters samples whose answers or reasoning traces are inconsistent with our annotations. For captioning, we employ a multi-criteria scoring scheme focusing on node coverage and edge accuracy, retaining only high-consistency samples. This process yields a 120K-scale corpus $\mathcal{D}_{SG}$ (with failure rate below 1% randomly cross-validated by human experts), establishing a robust foundation for graph-aligned reasoning. Detailed data composition is shown in Figure 3 (c).

### 3.4. Node-as-proxy Graph Reward

To internalize the graph-aligned reasoning skills, we first apply supervised fine-tuning with 120K generated data as cold start in Figure 4 (a). Subsequently, we introduce a GRPO-based reinforcement learning. We contend that defin-

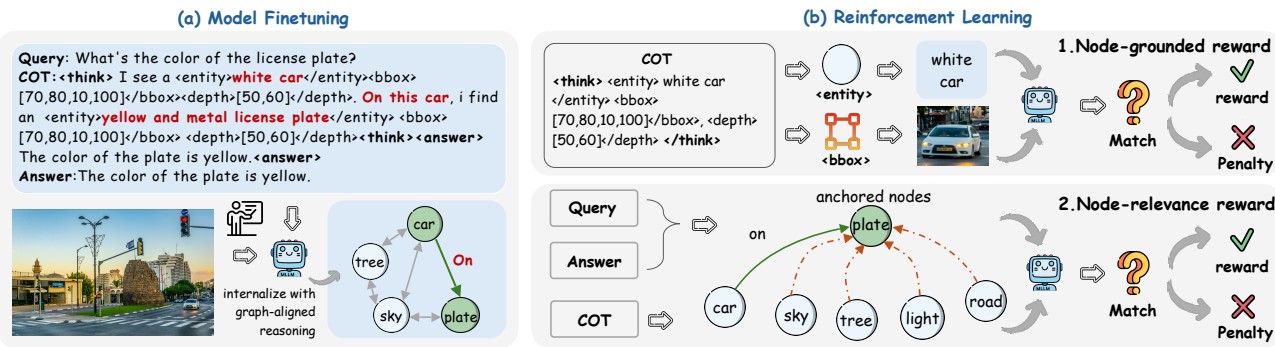

*Figure 4.* **Two-stage Training Pipeline.** We first adopt the (a) supervised fine-tuning stage to internalize the MLLMs with graph-aligned reasoning. Then (b) a reinforcement learning stage is conducted with proxy-as-node graph rewards, consolidating the structured reasoning.

ing a verifiable edge-level supervision signal is inherently ambiguous, as a single pair of nodes can be linked by multiple valid relationships. Instead of restricting reasoning traces through edge-level constraints, we observe that any valid trajectory inevitably traverses a sequence of visually grounded and semantically consistent anchor nodes. Motivated by this, we introduce a ***node-as-proxy graph reward*** scheme with two complementary rewards, treating nodes as proxies to explore valid navigation.

**Node-grounded Reward.** To ensure the correctness of node proxies along reasoning traces, we introduce a node-grounded reward that *aligns visual attention with the reasoning trace*. For each visited node $v_t$, we first extract its textual description $e_t$ from the `<entity>` tag. Given the bbox $b_t$, we then crop the corresponding visual region from $I_i$, denoted as $\phi(I_i, b_t)$. In Figure 4 (b), we employ an MLLM as a binary judge $g_{\text{vis}}(\cdot)$ to verify cross-modal alignment:

$$r_{\text{gro}}(v_t) = g_{\text{vis}}(e_t, \phi(I_i, b_t)). \quad (5)$$

A node receives a positive reward 1.0 only when the visually grounded region is consistent with the node descriptions.

**Node-relevance Reward.** Instead of constraining navigation with predefined edge sequences, we introduce a node-relevance reward. It *encourages the model to traverse nodes that **align with the query intent** and **reduces detours** caused by ambiguous graph structures*. To assess semantic relevance, we first adopt an LLM-as-judge to identify nodes explicitly mentioned in the query and answer, treating them as anchored nodes $\mathcal{A}$. During rollout, nodes matching these anchors receive *positive* rewards. Beyond these, visited nodes directly connected to the anchored are rewarded only when the LLM judge determines that they provide necessary semantic support for the target navigation. This design maintains flexible graph reasoning while preventing spurious rewards from loosely related nodes, effectively aligning navigation with the query's semantic intent. Formally, for each visited node $v_t$ with entity description $e_t$, we define

the node-relevance reward as:

$$r_{\text{rel}}(v_t) = \begin{cases} \mathcal{R}, & e_t \in \mathcal{A} \text{ or semantically supports } \mathcal{A}, \\ 0, & \text{otherwise}, \end{cases}$$

$$(6)$$

where $\mathcal{R}$ reflects various degrees of semantic relevance, with higher scores indicating stronger alignment with the query intent. Details are specified in Supplementary Materials.

**Overall Reward Design.** In addition to the node-proxy rewards, the total reward also includes an accuracy reward and a formatting reward. The accuracy reward checks whether the predicted answer matches the ground truth, while the formatting reward ensures the output follows the required structure, including both the reasoning format and the traced node specification. For a rollout with reasoning trajectory $\tau$, the total reward is defined as:

$$R(\tau) = \frac{1}{T} \sum_{t=1}^{T} r_{\text{gro}}(v_t) + \frac{1}{|\mathcal{A}|} \sum_{t=1}^{T} r_{\text{rel}}(v_t) + r_{\text{acc}} + r_{\text{fmt}}, \quad (7)$$

where $T$ is the number of visited nodes in $\tau$, $|\mathcal{A}|$ is the number of anchored nodes, and $r_{\text{acc}}$ and $r_{\text{fmt}}$ denote the answer accuracy and formatting rewards. With these rewards, we guide the model to learn reasoning trajectories that are both *semantically coherent* and *visually grounded*, strengthening its reasoning to follow the scene graph structure.

### 3.5. Model Optimization

We adopt SFT model as the reference $\pi_{\text{ref}}$ and use *Group Relative Policy Optimization* (GRPO) (Guo et al., 2025) to optimize the policy $\pi_\theta$. It compares rewards within a group of sampled rollouts, emphasizing relative advantages $\hat{A}$. The training objective is formulated as:

$$\mathcal{L}_{\text{GRPO}} = -\mathbb{E}\Big[ \min(\rho(\theta)\hat{A}, \text{clip}(\rho(\theta), 1 - \epsilon, 1 + \epsilon)\hat{A})\Big] + \beta \, \text{KL}(\pi_\theta \,\|\, \pi_{\text{ref}}),$$

$$(8)$$

where $\rho(\theta)$ is the probability ratio between the current and old policies, $\epsilon$ is the clipping parameter and $\beta$ is the coeffi-

*Table 1.* **Performance comparison on high-resolution benchmarks.** FSP means Fine-grained Single-instance Perception. FCP means Fine-grained Cross-instance Perception. Accuracy is used as the metric.

| Methods | Param Size | VStarBench | | | HRBench-4K | | | HRBench-8K | | |
| --- | --- | --- | --- | --- | --- | --- | --- | --- | --- | --- |
| | | Overall | Attribute | Spatial | Overall | FSP | FCP | Overall | FSP | FCP |
| GPT-4o (Hurst et al., 2024) | - | 67.5 | 72.2 | 60.5 | 65.0 | 66.8 | 63.3 | 59.6 | 60.8 | 58.5 |
| GPT-5-Mini (OpenAI, 2025) | - | 63.9 | - | - | 66.3 | - | - | 60.9 | - | - |
| Gemini-2.5-Pro (Google, 2025) | - | 79.2 | - | - | - | - | - | - | - | - |
| LLaVA-OV (Li et al., 2024) | 7B | 70.7 | 73.0 | 60.5 | 64.3 | 74.8 | 53.8 | 59.8 | 65.3 | 54.3 |
| LLaVA-OV (Li et al., 2024) | 72B | 73.8 | 80.9 | 63.2 | 66.3 | 76.5 | 56.0 | 60.9 | 68.8 | 53.0 |
| InternVL3 (Zhu et al., 2025a) | 8B | 72.3 | 73.0 | 71.1 | 70.8 | 79.3 | 62.3 | 62.0 | 64.3 | 59.8 |
| InternVL3 (Zhu et al., 2025a) | 78B | 76.4 | 75.7 | 77.6 | 75.5 | 84.5 | 66.5 | 67.3 | 71.8 | 62.8 |
| Qwen2.5-VL (Bai et al., 2025b) | 3B | 75.4 | 80.9 | 67.1 | 66.0 | 82.3 | 49.8 | 63.1 | 80.3 | 46.0 |
| Qwen2.5-VL (Bai et al., 2025b) | 7B | 76.4 | 77.4 | 75.0 | 68.8 | 85.2 | 52.2 | 65.3 | 78.8 | 51.8 |
| Qwen2.5-VL (Bai et al., 2025b) | 32B | 81.2 | 77.4 | 86.8 | 73.4 | 87.5 | 59.3 | 70.4 | 82.3 | 58.5 |
| SEAL (Wu & Xie, 2024) | 7B | 75.4 | 74.8 | 76.3 | - | - | - | - | - | - |
| DyFo (Li et al., 2025c) | 7B | 81.2 | 80.0 | 82.9 | - | - | - | - | - | - |
| Vision-R1 (Huang et al., 2025) | 7B | 79.6 | 80.0 | 79.0 | 74.0 | 88.5 | 59.5 | 69.6 | 83.3 | 56.0 |
| LVR (Li et al., 2025b) | 7B | 81.7 | 82.6 | 80.3 | 69.6 | 79.3 | 60.0 | 66.1 | 78.5 | 53.8 |
| Pixel-Reasoner (Wang et al., 2025b) | 7B | 80.6 | 83.5 | 76.3 | 72.9 | 86.0 | 60.3 | 66.9 | 80.0 | 54.3 |
| DeepEyes (Zheng et al., 2025) | 7B | 85.6 | 87.3 | 84.5 | 75.1 | 91.3 | 59.0 | 72.6 | 86.8 | 58.5 |
| Thyme (Zhang et al., 2025b) | 7B | 82.2 | 83.5 | 80.3 | 77.0 | 91.0 | 63.0 | 72.0 | 86.5 | 57.5 |
| **SaGe-3B (Ours)** | **3B** | 89.0 | 86.1 | 93.4 | 72.0 | 89.0 | 55.0 | 68.9 | 85.0 | 52.8 |
| **v.s. Qwen2.5-VL-3B** | **3B** | **+13.6** | **+5.2** | **+26.3** | **+6.0** | **+6.7** | **+5.2** | **+5.8** | **+4.7** | **+6.8** |
| **SaGe-7B (Ours)** | **7B** | 89.0 | 85.2 | 94.7 | 76.5 | 94.0 | 59.0 | 73.4 | 89.8 | 57.8 |
| **v.s. Qwen2.5-VL-7B** | **7B** | **+12.6** | **+7.8** | **+19.7** | **+7.7** | **+8.8** | **+6.7** | **+8.1** | **+11.0** | **+6.0** |

cient of KL divergence penalty.

## 4. Experiments and Results

### 4.1. Settings

**Benchmarks and Metrics.** We consider three types of MLLM benchmarks. (1) *Fine-grained perception* on high-resolution images, including VStarBench (V*) (Wu & Xie, 2024), HRBench-4K (Wang et al., 2025e), and HRBench-8K (Wang et al., 2025e). (2) *Spatial perception and reasoning*, including CVBench-2D (Tong et al., 2024), and CVBench-3D (Tong et al., 2024). (3) *General visual benchmarks*, covering general comprehension (MMStar (Chen et al., 2024)), grounding ability (RefCOCO (Kazemzadeh et al., 2014)), and chart understanding (ChartQA (Masry et al., 2022)). Accuracy and Acc@0.5 are the metrics for VQA and grounding tasks, respectively.

**Compared Baselines.** We compare SaGe with three categories of baselines. (1) Advanced proprietary models, such as GPT-4o (Hurst et al., 2024), Gemini-2.5-Pro (Google, 2025), and GPT-5-Mini (OpenAI, 2025). (2) State-of-the-art open-source models, such as Qwen2.5-VL (Bai et al., 2025b), Intern3VL (Zhu et al., 2025a), and LLaVA-OneVision (Li et al., 2024). (3) Advanced methods for fine-grained tasks, including tool-used or latent-reasoning approaches, such as DeepEyes (Zheng et al., 2025), Pixel-Reasoner (Wang et al., 2025b), and LVR (Li et al., 2025b).

**Implementation Details.** Qwen2.5-VL-3B and 7B are used as our baselines. 120K structured data in the cold-start stage is mainly generated from SA-1B (Kirillov et al., 2023). In the RL stage, we use 26K samples, including 2K grounding data from (Lin et al., 2014), 4K chart data from (Masry et al., 2022), and 20K samples from V* train set following (Zheng et al., 2025). More details are in Supplementary Materials.

### 4.2. Main Results

**Performance of Fine-grained Perception.** With our data and node-as-proxy rewards, SaGe yields substantial gains across three high-resolution benchmarks. As shown in Table 1, SaGe improves Qwen2.5-VL-3B and Qwen2.5-VL-7B on VStarBench, from 75.4 and 76.4 to 89.0 and 89.0, respectively. The consistent gains are shown on HRBench-4K and HRBench-8K as well, attributed to our hierarchical perception and node evidence for reasoning. Notably, with only 3B parameters, SaGe-3B outperforms the proprietary model GPT-4o and remains competitive with much larger model InternVL3-78B. Furthermore, SaGe-7B clearly outperforms Qwen2.5-VL-32B as well as tool-based method (Zheng et al., 2025) with complex inference pipelines.

**Performance of Spatial Understanding.** Each node is articulated with bbox and depth cues, enabling strong spatial understanding. As shown in Table 2, SaGe-3B achieves scores of 77.8 and 72.3 on CVBench-2D and CVBench-3D, outperforming Qwen2.5-VL-3B by 10.8% and 5.8%,

*Table 2.* **Performance comparison on spatial understanding and general vision benchmarks.**

| Methods | Param Size | CVBench-2D | | | CVBench-3D | | | Other Tasks | | |
|---|---|---|---|---|---|---|---|---|---|---|
| | | Overall | Count | Relation | Overall | Depth | Distance | MMstar | RefCOCO | ChartQA |
| GPT-4o (Hurst et al., 2024) | - | 67.5 | 72.2 | 60.5 | 65.0 | 66.8 | 63.3 | 65.7 | - | - |
| LLaVA-OV (Li et al., 2024) | 7B | 73.8 | 67.9 | 79.8 | 77.2 | 83.8 | 70.5 | 61.7 | - | 80.0 |
| Qwen2.5-VL (Bai et al., 2025b) | 3B | 67.0 | 62.4 | 72.6 | 66.5 | 73.6 | 59.3 | 52.2 | 86.3 | 82.2 |
| Qwen2.5-VL (Bai et al., 2025b) | 7B | 73.6 | 62.4 | 84.9 | 73.3 | 79.8 | 66.7 | 60.3 | 88.1 | 85.6 |
| Qwen2.5-VL (Bai et al., 2025b) | 32B | 76.2 | 63.2 | 92.0 | 83.1 | 84.3 | 81.8 | 68.4 | 92.7 | 88.1 |
| LVR (Li et al., 2025b) | 7B | 73.4 | 62.4 | 86.8 | 77.6 | 82.0 | 73.2 | 61.1 | 73.8 | 76.9 |
| Vision-R1 (Huang et al., 2025) | 7B | 71.1 | 60.9 | 83.5 | 76.3 | 78.8 | 73.8 | 61.8 | 72.5 | - |
| **SaGe-3B (Ours)** | **3B** | 77.8 | 67.9 | 89.9 | 72.3 | 79.2 | 65.3 | 54.4 | 89.0 | 83.8 |
| **v.s. Qwen2.5-VL-3B** | **3B** | **+10.8** | **+5.5** | **+17.3** | **+5.8** | **+5.6** | **+6.0** | **+2.2** | **+2.7** | **+1.6** |
| **SaGe-7B (Ours)** | **7B** | 79.4 | 68.0 | 93.2 | 80.5 | 88.8 | 72.2 | 62.3 | 89.8 | 87.2 |
| **v.s. Qwen2.5-VL-7B** | **7B** | **+5.8** | **+5.6** | **+8.3** | **+7.2** | **+9.0** | **+5.5** | **+2.0** | **+1.7** | **+1.6** |

**Coarse Perception:** 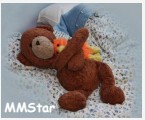
**Question: Which is the main topic of the image?**
Qwen2.5-VL-3B: **The toy bear is the main topic** in the image.
SaGe (Ours): Thinking: I observe a <entity>toy bear</entity><bbox>[[56,73,439,362]]</bbox><depth>[[108,254]]</depth> and a <entity>toy chicken</entity><bbox>[[177,116,356,199]]</bbox><depth>[[107,150]]</depth>. The toy bear and chicken are clearly visible in the image. Output: **A toy bear and a toy chicken.**

**Fine-grained Perception:** 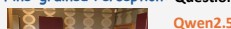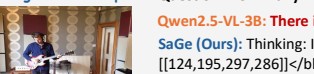
**Question: How many musical instruments are in the image?**
Qwen2.5-VL-3B: **There is one guitar** in the image.
SaGe (Ours): Thinking: I observe a <entity>guitar</entity><bbox>[[124,195,297,286]]</bbox><depth>[[30,100]]</depth> and a <entity>second guitar</entity><bbox>[[203,246,250,381]]</bbox><depth>[[20,30]]</depth>. Both are distinct instruments, making a total of two. Output: **There are two instruments in the image.**

**2D Understanding:** 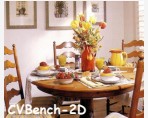
**Question: Is the jar left or right to the pitcher?**
Qwen2.5-VL-3B: **The jar is on the right side of the pitcher.**
SaGe (Ours): Thinking:I observe a <entity>jar</entity><bbox>[[564,270,671,450]]</bbox><depth>[[39,58]]</depth> and a <entity>pitcher</entity><bbox>[[770,306,908,424]] </bbox><depth>[[37,46]]</depth>. **The jar's x-range (564-671) lies entirely left of the pitcher's x-range (770-908).** Therefore the jar is on the left side of the pitcher. Output: **The jar is on the left side of the pitcher.**

**3D Understanding:** 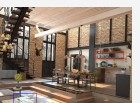
**Question: Are the shelves or the books closer to the camera ?**
Qwen2.5-VL-3B: **The shelves are closer to the camera.**
SaGe (Ours): Thinking: I observe a <entity>shelves</entity><bbox>[[759,286,971,346]]</bbox><depth>[[30,37]]</depth> and a <entity>books</entity><bbox>[[182,580,324,646]]</bbox><depth>[[166,200]]</depth>. **The books have a much larger depth range (166-200) than the shelves (30-37), meaning they are nearer to the foreground.** Therefore the books are closer. Output: **The books are closer.**

**High Resolution:** 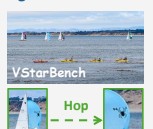
**Question: What kind of animal is on the blue sail?**
Qwen2.5-VL-3B: **A bird is on the blue sail.**
SaGe (Ours): Thinking: I first observe a <entity>blue sail</entity><bbox>[[790,386,915,579]]</bbox><depth>[[36,40]]</depth>. **Attached to it,** a <entity>spider</entity><bbox>[[846,437,912,550]]</bbox><depth>[[36,40]]</depth>, is printed on the sail. Therefore, the animal on the blue sail is a spider. Output: **There is a spider on the blue sail.**

**Chart/Bar:** 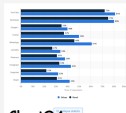
**Question: What is the difference between highest and lowest value of dark blue bar?**
Qwen2.5-VL-3B: **48%.**
SaGe (Ours): I can see a <entity>lowest dark blue bar</entity><bbox>[[69,385,240,447]]</bbox><depth>[[107,115]]</depth> with a value of 22%. The <entity>highest dark blue bar</entity><bbox>[[69,30, 690,100]]</bbox><depth>[[107,115]]</depth>has a value of 75%. **The difference is 75% - 22% = 53%.** Output: **53%.**

*Figure 5.* **Case study on various QA scenarios.** Compared to baseline, our method generates correct answers with node-articulated CoT.

*Table 3.* **Ablation study of key training stages.** SaGe-3B is used. V*: VStarBench. CV-2D: CVBench-2D. CV-3D: CVBench-3D.

| Condition | SFT | GRPO | NPR | V* | CV-2D | CV-3D |
|---|---|---|---|---|---|---|
| Qwen2.5-VL-3B | | | | 75.4 | 67.0 | 66.5 |
| w/ Coldstart | ✓ | | | 83.2 | 75.5 | 69.7 |
| w/ GRPO | ✓ | ✓ | | 85.9 | 76.2 | 69.3 |
| **SaGe (Ours)** | ✓ | ✓ | ✓ | **89.0** | **77.8** | **72.3** |

respectively. For 2D spatial relation, SaGe-3B boosts the baseline from 72.6 to 89.9, achieving a 17.3% improvement. SaGe-7B surpasses all comparable baselines, reaching 79.4 on CVBench-2D. Furthermore, with explicit depth modeling in our CoT, SaGe-7B even outperforms Qwen2.5-VL-32B, showing strong depth-aware spatial reasoning capability.

**Performance of General Vision Tasks.** While optimized for fine-grained reasoning, **SaGe** exhibits generalization across diverse vision benchmarks, including understanding (MMStar), grounding (RefCOCO), and chart analysis (ChartQA). As shown in Table 2, SaGe-7B achieves 62.3% accuracy on MMStar. Notably, it delivers stronger ground-

ing performance on the RefCOCO val, surpassing the baseline by 1.7%. Similar improvements are observed in chart understanding, with a 1.6% gain on ChartQA. We attribute these gains to the hierarchical perception and node-level rewards, promoting precise localization and node navigation.

**Visualized Structure QA Cases.** As shown in Figure 5, we present qualitative examples across diverse QA scenarios. Compared to baseline, SaGe demonstrates stronger visual performance on both coarse and fine-grained perception tasks. Notably, SaGe performs multi-hop reasoning and successfully navigates to small target entities via intermediate nodes in challenging high-resolution case, highlighting its hierarchical perception and structured reasoning.

### 4.3. Ablation Studies

**Effectiveness of Training Stages.** Table 3 reports ablations over different training stages with Qwen2.5-VL-3B as the baseline. Leveraging graph-aligned structured data, cold-start SaGe-3B already achieves clear gains on VS-

*Table 4.* **Ablation study of our CoT parterns.** SaGe-3B is used.

| Condition | <entity> | <bbox> | <depth> | V* | CV-2D | CV-3D |
|---|---|---|---|---|---|---|
| Answer-only | | | | 79.1 | 72.8 | 66.4 |
| w/ <entity> | ✓ | | | 79.9 | 72.3 | 65.7 |
| w/ <bbox> | ✓ | ✓ | | 80.6 | 75.0 | 66.1 |
| **SaGe (Ours)** | ✓ | ✓ | ✓ | **83.2** | **75.5** | **69.7** |

*Table 5.* **Ablation of node-as-proxy rewards.** SaGe-3B is used.

| Condition | GRPO | NGR | NRR | V* | CV-2D | CV-3D |
|---|---|---|---|---|---|---|
| GRPO | ✓ | | | 85.9 | 76.2 | 69.3 |
| w/o NGR | ✓ | | ✓ | 87.1 | 76.9 | 71.5 |
| w/o NRR | ✓ | ✓ | | 85.0 | 74.6 | 67.6 |
| **SaGe (Ours)** | ✓ | ✓ | ✓ | **89.0** | **77.8** | **72.3** |

*Table 6.* **Comparison with the teacher model Qwen2.5-VL-72B.**

| Methods | V* | | CV-2D | | CV-3D | |
|---|---|---|---|---|---|---|
| | Attr. | Spat. | Count | Rela. | Depth | Dist. |
| Qwen2.5-VL-72B | 79.1 | 89.5 | 67.3 | 93.1 | 87.9 | **83.8** |
| **SaGe-7B (Ours)** | **85.2** | **94.7** | **68.0** | **93.2** | **88.8** | 72.2 |

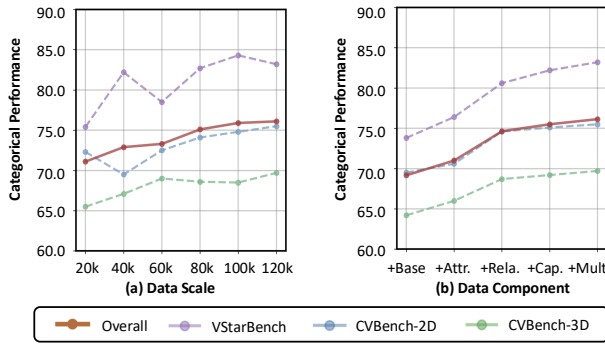

*Figure 6.* **Data Scaling and Composition Analysis.** Base: grounding and counting. Cap.: caption. Mult.: multi-hop reasoning data.

tarBench (V*) and CVBench, demonstrating that scene-graph–aligned supervision effectively injects structured visual knowledge. Introducing a GRPO-based RL stage with accuracy and format rewards further improves V* to 85.9 by encouraging exploration of diverse reasoning trajectories. Finally, incorporating the node-as-proxy reward (NPR) yields a substantial boost to 89.0, as it jointly enforces node-level visual–textual grounding and constrains the reasoning search space, enabling efficient navigation toward target-relevant nodes while suppressing irrelevant exploration.

**Effectiveness of Node-articulated CoT.** Our data features node-articulated CoT, where each node provides reasoning evidence. As reported in Table 4, we evaluate different reasoning-trace designs. Answer-only excludes CoT from the 120K curated data and uses the 3B model as baseline, achieving 79.1 on V*. Introducing CoT with node descriptions tagged by <entity> improves performance to 79.9, highlighting the benefit of explicit attribute reference. Further incorporating <bbox> injects localization into the reasoning trace, providing 2D spatial evidence and yielding 2.7% gain on CVBench-2D. When <depth> cues are further added, the model gains explicit 3D spatial awareness, boosting CVBench-3D performance to 69.7. These results validate the efficacy of our structured CoT.

**Effectiveness of Node-as-proxy Rewards.** In Table 5, we evaluate the efficacy of our node-as-proxy rewards against vanilla GRPO. Introducing the node-relevance reward (NRR) improves V* from 85.9 to 87.1, demonstrating it regularizes reasoning traces by encouraging traversal over target-relevant nodes while suppressing irrelevant ones. The node-grounded reward (NGR) enforces node-level visual–textual alignment. However, when applied alone, it consistently degrades performance, indicating that node-

level supervision without trace-level guidance may lead to reward hacking, where the model attends to visually salient but target-irrelevant nodes. ***When combined, NRR and NGR exhibit strong complementarity: NRR guides which nodes to traverse, while NGR ensures the correctness of the visited nodes.*** This synergy further boosts V* performance, validating the design of our reward scheme.

### 4.4. Further Analysis

**Comparison with Teacher Model.** We construct scene graphs through a hybrid pipeline leveraging Qwen2.5-VL-72B. Although our pipeline is primarily built upon this model, SaGe-7B consistently surpasses its teacher on V*, CVBench-2D, and depth benchmarks, as evidenced in Table 6. This performance gain is attributed to our cascaded pipeline, which explicitly mines hierarchical entities, precise 2D localizations, depth-aware semantics, and diverse relations, effectively addressing the structured knowledge deficiencies inherent in Qwen2.5-VL-72B. These results underscore SaGe's ability to extract hierarchical knowledge, transcending the capabilities of the original teacher model.

**Data Scaling and Component Analysis.** As shown in Figure 6, we analyze the impact of different training data scale and composition. In Figure 6 (a), we investigate the effect of training data scale. As the scale of training data increases, the overall performance cross different benchmarks consistently improves. It validates the effectiveness and scalability of our structured data generated from scene graph. In Figure 6 (b), SaGe exhibits steady performance gains as different types of graph-aligned training data are progressively introduced. In particular, incorporating multi-hop reasoning traces and graph-traversal captions enables SaGe to internalize hierarchical reasoning patterns, further boosting performance from 80.6 to 83.2 on VStarBench.

## 5. Conclusion

In this paper, we propose Scene Graph Thinking (SaGe), a novel paradigm that equips MLLMs with fine-grained and

structured visual reasoning through scene-graph representations. Specifically, we first introduce an automated data engine that transforms flat image–text pairs into hierarchical scene graphs. Then training data are sampled from graphs to internalize structured reasoning. Furthermore, we develop node-as-proxy graph rewards for reinforcement finetuning, enabling efficient target navigation over scene graphs. With curated data and graph-aligned training, SaGe improves performance across eight visually intensive benchmarks.

## Impact Statement

This paper presents work whose goal is to advance the field of Machine Learning. There are many potential societal consequences of our work, none of which we feel must be specifically highlighted here.

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

# Supplementary Materials of Scene Graph Thinking

In this appendix, we provide detailed implementation specifications for Scene Graph Thinking (SaGe), along with supplementary analyses and discussions that extend the main paper.

- Appendix A: Additional implementation details, including training configurations and optimization settings of SaGe.

- Appendix B: Supplementary results, including visualized QA cases and extended discussions on model generalization.

- Appendix C: Data construction pipeline and detailed composition of the large-scale structured training corpus.

- Appendix D: LLM-as-Judge prompts and reward design for node-as-proxy graph reasoning.

## A. Implementation Details

We store each scene graph in a JSON format, where every node in an image—including its category name, attributes, bounding box, depth cues, and relations to other nodes—is explicitly represented. During sampling, the stored scene graph is used as structured graph knowledge to guide our LLM-based data construction process.

We adopt Qwen2.5-VL-3B and Qwen2.5-VL-7B as our baseline models. During the cold-start stage, the models are trained for one epoch on 120K collected structured samples. The majority of the generated data are sourced from SA-1B (Kirillov et al., 2023), whose high-resolution images and rich object annotations are particularly well suited for fine-grained visual and relational analysis. The remaining 10K counting samples are collected from COCO 2014 (Lin et al., 2014). Random scaling is adopted for data augmentation. The learning rate is set to $1e-5$ with a total batch size of 256.

In the RL post-training stage, we use a total of 26K samples, including 2K grounding samples from (Lin et al., 2014), 4K chart-related samples from (Masry et al., 2022), and 20K samples from V* following (Zheng et al., 2025). We adopt Qwen3-VL-30B-A3B-Instruct (Bai et al., 2025a) as the judge for node-grounded rewards, while Qwen3-30B-A3B-Instruct (Yang et al., 2025a) is used as the judge for both node-relevance rewards and accuracy rewards. For multiple-choice QA tasks, the accuracy reward is set to 1 if the predicted answer matches the ground truth, and 0 otherwise. For grounding-type data, we compute the IoU between the predicted bounding box and the ground-truth box, which is directly used as the reward signal. The learning rate for RL training is set to $1e-6$ with a total batch size of 128. The model is trained for one epoch, with the number of rollouts set to 8.

## B. More Results and Discussions

### B.1. Quality of Scene Graphs

To assess scene graph quality, we evaluate 1,000 sampled raw graphs using Gemini-3-Pro (Google DeepMind, 2025), measuring the correctness of 22029 nodes and 26283 edges. A node is considered correct if its bounding box region matches its entity tag and attributes. It reports that the nodes and edges in our constructed graph exhibit high precision, with 94.5% of nodes and 92.9% of edges correctly identified. Regarding node recall and depth, exhaustive node annotation is prohibitively challenging due to the open-ended and hierarchical nature of scene graphs. Our depth signals provide range cues for depth reasoning rather than precise estimation, and their effectiveness is supported by strong performance on the depth-related benchmark CVBench-3D (Table 2 in the main text). We also provide error breakdowns in Figure 7 and Figure 8, where 52.0% of node errors arise from mismatches between node attributes and visual regions, and 55.6% of edge errors stem from invalid relations. In our data construction pipeline, we employ a two-round verification process, consisting of a GME-based filtering step followed by MLLM re-verification during QA construction. Based on the generated high-quality scene graphs and this verification process, we obtain curated data with a failure rate of less than 1% under random human cross-validation.

## Node Error Breakdown
### Total Nodes:22029| Errors:1222| Accuracy:94.5%

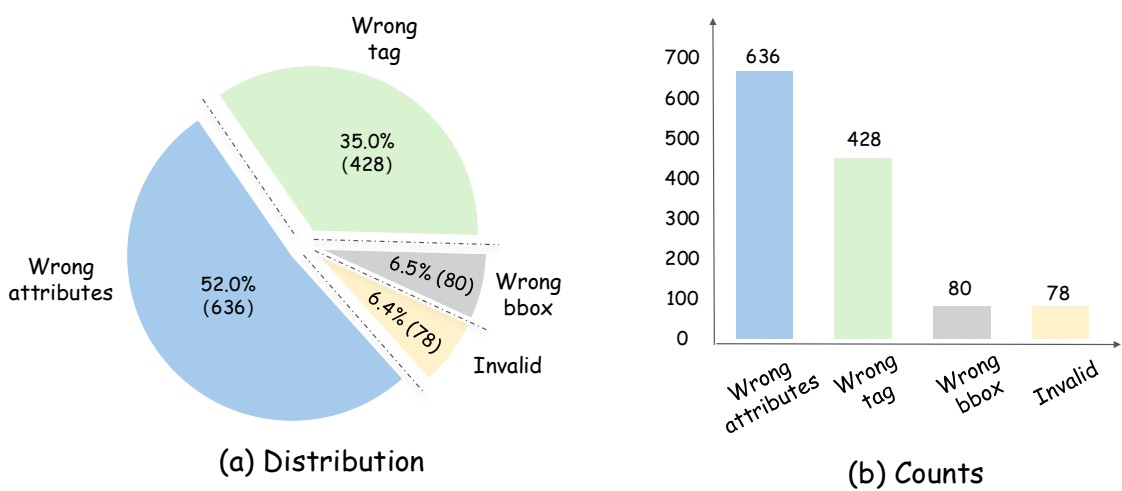

*Figure 7.* **Node Error Breakdown.** (a) Distribution of node errors. (b) Number distribution of different error types. The nodes constructed in our scene graph achieve an accuracy of 94.5%, demonstrating the high quality of our scene graph.

## Edge Error Breakdown
### Total Edges:26283 | Errors:1874 | Accuracy:92.9%

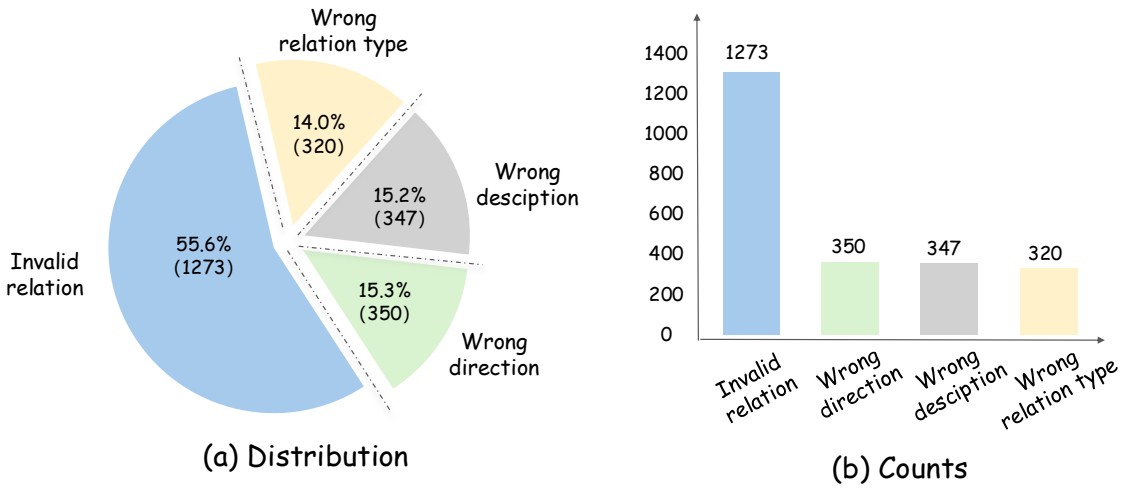

*Figure 8.* **Edge Error Breakdown.** (a) Distribution of edeg errors. (b) Number distribution of different error types. The edges in our constructed scene graph attain an accuracy of 92.9%, which validates the reliability and quality of the scene graph construction.

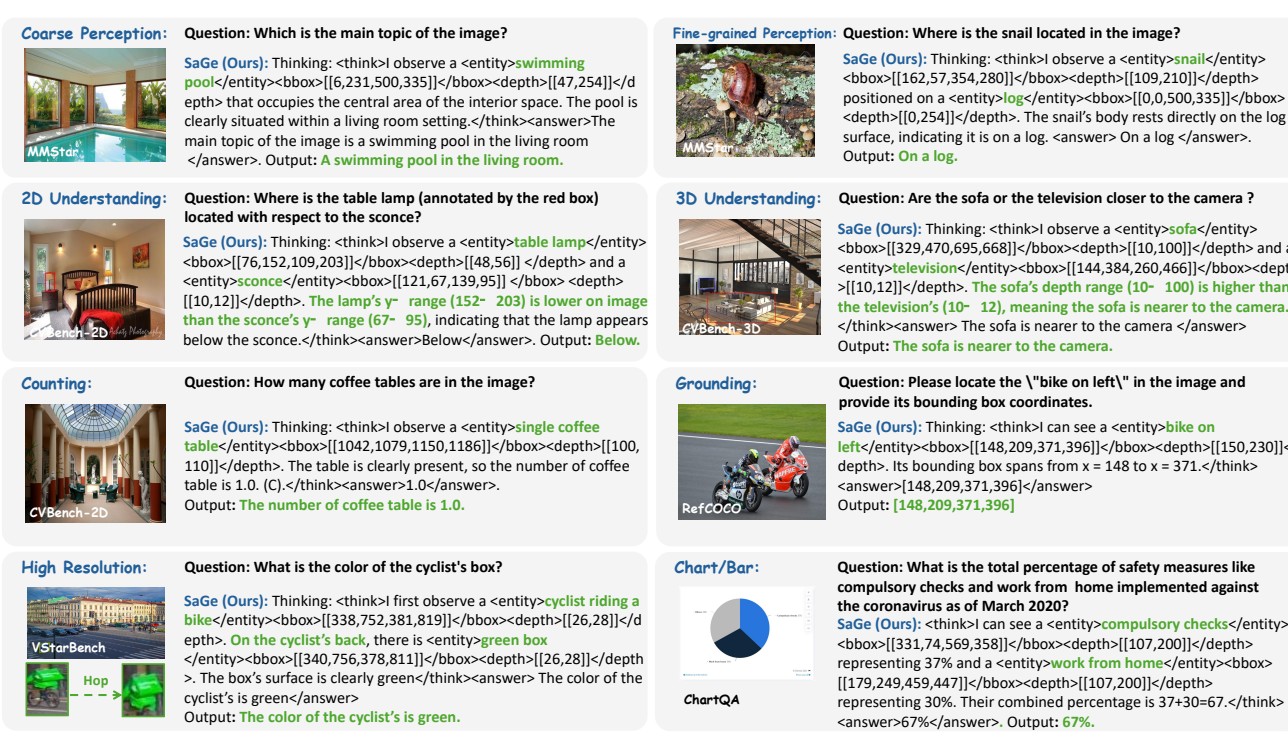

*Figure 9.* **Case study on various QA scenarios.** Our method demonstrates strong spatial understanding and fine-grained perception through node-articulated CoT, highlighting the effectiveness of our scene-graph–based structured reasoning.

### B.2. More Visualized QA Cases

As illustrated in Figure 9, we visualize additional QA cases to further demonstrate the effectiveness of our method. SaGe consistently exhibits stronger visual understanding and reasoning performance across a variety of tasks.

## C. Details for Constructed Data

### C.1. Data Components

To cultivate scene-graph–aligned reasoning, we develop an automated data engine to construct a large-scale training corpus consisting of 120K samples. The corpus covers diverse forms of VQA pairs and captioning data. Specifically, it includes 25K attribute-centric samples, comprising 10K color-related, 5K state-related, 5K shape-related, and 5K material-related instances. The distribution of attribute types used for SFT training is illustrated in Figure 10. As shown, the `<entity>` tag in our CoT explicitly wraps each node name together with its attributes, providing attribute-level clues that facilitate structured reasoning.

In addition, the corpus contains 30K spatial reasoning samples, covering both 2D and 3D spatial reasoning tasks. As illustrated in Figure 11, the 2D data focus on planar spatial relations such as left/right and above/below, while the 3D data involve front/behind localization queries. The `<bbox>` tag in our CoT encodes the spatial coordinates of entities, providing explicit positional evidence for 2D spatial reasoning. Meanwhile, the `<depth>` tag captures depth-related clues, supplying front–back evidence to support 3D spatial reasoning.

The corpus further incorporates 25K samples with multi-hop reasoning traces, together with 20K caption samples. The caption data are further divided into 10K local captions and 10K global captions. As shown in Figure 12, local captions focus on traversing nodes within specific regions, such as a bounding box or the half of an image, and provide fine-grained regional descriptions. In contrast, global captions require the model to traverse all nodes in the scene graph and generate a holistic description of the entire image, as illustrated in Figure 13. This data enables effective traversal of relational traces toward target nodes during reasoning.

To further enhance data diversity, we additionally generate 10K grounding samples and 10K counting samples. Representative

examples of the constructed data are presented in Figure 11.

## C.2. Caption Filtering

To ensure the quality of the generated captions, we employ a multi-criteria scoring scheme that emphasizes node coverage and edge accuracy, retaining only high-consistency samples. The multi-criteria scoring scheme is illustrated in Figure 14. Specifically, the scheme comprises eight evaluation criteria that jointly assess the generated captions from multiple aspects, including a human-like descriptive style (persona-consistent reasoning without machine-centric phrasing), strict entity formatting and tag completeness (consistent use and ordering of `<entity>`, `<bbox>`, and `<depth>`), proper grouping of multiple instances, logical efficiency without redundant descriptions, faithful adherence to the user query, numerical validity of bounding boxes, and the avoidance of programmatic instance identifiers. Each satisfied criterion contributes one point, and only captions that meet all criteria are retained in the final dataset.

# D. Details for Node-as-proxy Rewards

## D.1. Prompts for LLM-as-Judge

To internalize graph-aligned reasoning capabilities, we introduce a ***node-as-proxy graph reward*** scheme composed of two complementary rewards, in which graph nodes are treated as proxies for valid navigation paths. To provide fine-grained, node-level supervision signals, we adopt an LLM-as-judge paradigm, leveraging MLLMs/LLMs as discriminative evaluators. Specifically, the node-grounded reward employs a multimodal large language model (MLLM) as the judge, while both the node-relevance reward and the accuracy reward adopt a large language model (LLM) as the judge, together offering dense and informative supervision for reasoning traces.

The prompt used for the node-grounded reward is provided in Figure 15. It requires the MLLM judge to strictly assess whether the visual region attended by a node is consistent with its corresponding node name and attribute descriptions, thereby ensuring node-level visual–textual alignment. The prompt designed for the node-relevance reward is also illustrated in Figure 16. It instructs the LLM judge to first determine the number of anchored nodes and then evaluate the relevance of the visited nodes with respect to them, encouraging the reasoning process to efficiently navigate toward target nodes. ***When combined, the node-relevance reward (NRR) and node-grounded reward (NGR) exhibit strong complementarity: NRR guides which nodes to traverse, while NGR verifies the correctness of the visited nodes***. Together, they effectively promote graph-aligned visual reasoning.

## D.2. Hierarchical Relevance Rewarding

In the node-relevance reward scheme, we assign reward scores hierarchically according to different degrees of semantic relevance, gently encouraging the model to prioritize semantically related nodes along the reasoning traces. As illustrated in Figure 16, the scheme defines five discrete relevance levels: 1.0 for exact, query-consistent matches between entity names and key attributes; 0.8 for semantically similar but non-identical matches; 0.5 for cases where the correct subject is identified but query-specific attributes are missing or misused; 0.2 for incorrect subject selection (e.g., shifts to a container, holder, or part); and 0.0 for conflicting, negated, or irrelevant entities. This multi-level scoring scheme gently encourages the model to prioritize semantically related nodes along the reasoning traces, mitigating the brittleness and sparse feedback issues inherent in binary reward formulations, and providing smoother and more informative supervision for graph-aligned exploration.

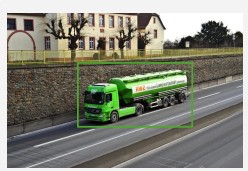

### Data Type: Color #10K

**Question: What is the color of the tanker truck? (A) green (B) gray (C) beige (D) black**
**Answer with the option's letter from the given choices directly.**

**Answer: A.**

**Thinking:** <think>I can see a <entity>**green tanker truck**</entity><bbox>[[262,191,625,375]]</bbox><depth>[[60,137]]</depth>. Its visible paint hue is a vivid green. Among the options, (A) green matches the observed color.</think><answer>A</answer>

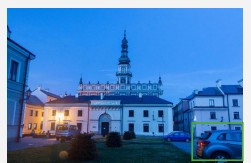

### Data Type: State #5K

**Question: What is the current state of the silver SUV visible in the foreground?**
**(A) moving (B) parked (C)   crashed (D) idle**
**Answer with the option's letter from the given choices directly.**

**Answer: B.**

**Thinking:** <think>I see a <entity>silver SUV parked in the foreground</entity><bbox>[[1554,996,1949,1276]]</bbox><depth>[[175,241]]</depth>. The vehicle is stationary and not in motion, indicating it is parked. Thus, option (B) is the correct state.</think><answer>B</answer>

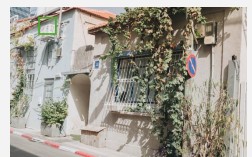

### Data Type: Shape #5K

**Question: What is the shape of the air conditioner unit attached to the building?**
**(A) round (B) rectangular (C) oval (D) triangular**
**Answer with the option's letter from the given choices directly.**

**Answer: B.**

**Thinking:** <think>I see a <entity>white rectangular air conditioner</entity><bbox>[[80,45,124,88]]</bbox><depth>[[77,92]]</depth>. The unit has a clear rectangular form, which aligns with option (B). Therefore the correct answer is (B).</think><answer>B</answer>

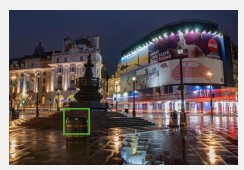

### Data Type: Material #5K

**Question: What material composes the body of the trash can that is positioned in front of the monument? (A) metal (B) wood (C) plastic (D) fabric**
**Answer with the option's letter from the given choices directly.**

**Answer: A.**

**Thinking:** <think>A <entity>trash can</entity><bbox>[[438,831,618,1020]]</bbox><depth>[[123,135]]</depth> stands in front of the monument. Its construction appears robust and metallic, signifying a metal composition. Option (A) metal is the correct selection.</think><answer>A</answer>

*Figure 10.* **Illustration of attribute-centric samples.** We generate 25K attribute-centric samples, comprising 10K color-related, 5K state-related, 5K shape-related, and 5K material-related instances. Each sample is accompanied by a node-articulated CoT, providing explicit evidence to support structured reasoning.

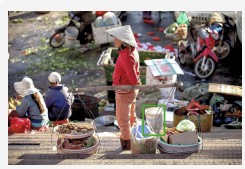

### Data Type: Grounding #10K

**Question: Please locate the cylindrical container made of light-colored material, possibly plastic or ceramic.**
**Respond with only a Python list of coordinates like `[x1, y1, x2, y2]`.**
**Answer: [616, 438, 702, 546].**

**Thinking:** <think>The user is asking for the location of a cylindrical container made of light-colored material, possibly plastic or ceramic. I can see <entity>a cylindrical container made of light-colored material, possibly plastic or ceramic</entity><bbox>[[616, 438, 702, 546]]</bbox><depth>[[56, 83]]</depth> located near the center-right of the scene, just above a basket and beside a package with floral wrapping. Its shape is clearly cylindrical with a smooth lid on top, matching the description. Since the user requested a Python list of coordinates, I will output the bounding box in the exact format specified.</think><answer>[616, 438, 702, 546]</answer>

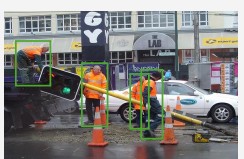

### Data Type: Counting #10K

**Question: How many people are there?**
**(A) 3 (B) 0 (C) 4 (D) 5**

**Answer: C.**

**Thinking:** <think>From this image, it appears there are 4 <entity>people</entity><bbox>[[295, 142, 349, 314], [271, 159, 304, 284], [167, 134, 224, 276], [23, 79, 99, 177]]</bbox><depth>[[20, 83], [20, 144], [1, 164], [3, 158]]</depth>. This leads to the correct answer as C.</think><answer>C</answer>

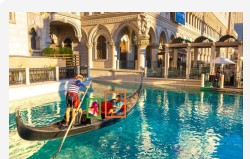

### Data Type: Spatial-2D #30K

**Question: Is the woman with the blue dress on the left or right of the woman with pink hair? (A) left (B) right**
**Answer with the option's letter from the given choices directly.**
**Answer: B.**

**Thinking:** <think>I observe a <entity>woman with pink hair</entity><bbox>[[294,343,352,413]]</bbox><depth>[[130,171]]</depth> and a <entity>woman in a blue dress</entity><bbox>[[374,318,434,404]]</bbox><depth>[[120,157]]</depth>. The blue-dress woman's x-range (374-434) is right of the pink-hair woman's x-range (294-352), so she is on the right. The correct answer is (B).<answer>B</answer></think>

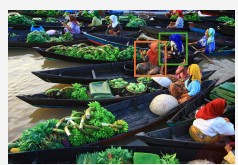

### Data Type: Spatial-3D #30K

**Question: Is the woman wearing a blue headscarf in front of or behind the woman wearing an orange headscarf? (A) in front (B) behind**
**Answer with the option's letter from the given choices directly.**
**Answer: B.**

**Thinking:** <think>I can see a <entity>woman wearing a blue headscarf</entity><bbox>[[958,139,1083,300]]</bbox><depth>[[45,59]]</depth> and a <entity>woman wearing an orange headscarf</entity><bbox>[[782,202,968,415]]</bbox><depth>[[52,92]]</depth>. The blue-headscarf woman's maximum depth (59) is lower than the orange-headscarf woman's minimum depth (52), meaning the orange-headscarf woman is closer to the foreground. Therefore the blue-headscarf woman is behind the orange-headscarf woman. The correct answer is (B).\n</think><answer>B</answer>

*Figure 11.* **Illustration of spatial reasoning, grounding, and counting samples.** We include 30K spatial reasoning samples, covering both 2D and 3D spatial reasoning tasks. The traceable evidence within the `<bbox>` and `<depth>` tags provides spatial clues to support reasoning. To further enhance data diversity, we additionally generate 10K grounding samples and 10K counting samples.

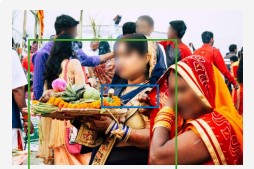

### Data Type: Multi-hop Navigate #25K

**Question: What is the color of the necklace with intricate designs?**

**Answer: The color of the necklace with intricate designs is golden.**

**Thinking:** </think> I see a <entity>**woman dressed in a blue and gold sari, adorned with jewelry and henna tattoos, carrying a tray filled with fruits and flowers**</entity><bbox>[[357, 147, 954, 959]]</bbox><depth>[[56, 162]]</depth>. Now, **focusing on her belongings**, I check her upper body for a necklace. A <entity>**gold necklace with intricate designs and a shiny texture**</entity><bbox>[[678, 424, 854, 516]]</bbox><depth>[[82, 90]]</depth>. Therefore, the color of the necklace with intricate designs is golden. </think><answer> The color of the necklace with intricate designs is golden.< answer>

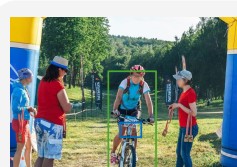

### Data Type: Multi-hop Navigate #25K

**Question: What is the number on the number plate with black text?**

**Answer: The number on the plate is '2114'.**

**Thinking:** <think>I first see a <entity>**woman wearing a blue and white cycling outfit, a pink helmet, and riding a bicycle**</entity><bbox>[[756, 366, 1099, 1141]]</bbox><depth>[[78, 180]]</depth>. Now, I'll look for the number plate as a detail **on this cyclist. Checking the sub-entities**, I find <entity>**a white number plate with black text displaying '2114' and additional text in a non-Latin script**</entity><bbox>[[867, 750, 974, 843]]</bbox><depth>[[166, 210]]</depth>. Therefore, the number of on the plate is '2114'.</think><answer> The number on the plate is '2114'</answer>

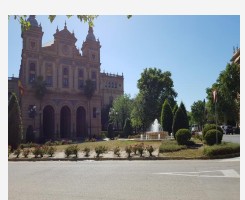

### Data Type: Local Caption #10K

**Question: Please provide a comprehensive visual description of the area within the coordinates [0,0,460,350].**

**Answer: Within the specified region, the central structure is a building. This building is characterized by two towers rising on its front façade, a series of windows arranged horizontally, an arch spanning the central entrance, a set of balconies above the upper story, two columns supporting the façade, and a covering of tiles that form the roof surface. Together these elements compose the historic edifice dominating the left side of the view.**

**Thinking:** <think>Within the specified region, the central structure is a <entity>**building**</entity><bbox>[[0,0,446,339]]</bbox><depth>[[24,48]]</depth>. This building is characterized by two <entity>**towers**</entity><bbox>[[48,0,140,159],[283,8,365,152]]</bbox><depth>[[37,43],[29,35]]</depth> rising on its front façade, a series of <entity>**windows**</entity><bbox>[[82,122,107,181],[149,131,172,188],[214,133,235,189],[276,133,297,189],[325,133,346,189]]</bbox><depth>[[39,41],[37,39],[36,37],[34,35],[32,34]]</depth> arranged horizontally, an <entity>**arch**</entity><bbox>[[136,233,312,330]]</bbox><depth>[[29,41]]</depth> spanning the central entrance, a set of <entity>**balconies**</entity><bbox>[[364,137,451,168]]</bbox><depth>[[18,23]] </depth>above the upper story, two <entity>**columns**</entity><bbox>[[176,224,200,325],[245,224,268,325]]</bbox><depth>[[36,40],[33,38]]</depth> supporting the façade, and a covering of <entity>**tiles**</entity><bbox>[[60,55,354,194]]</bbox><depth>[[27,46]]</depth> that form the roof surface. **Together these elements compose the historic edifice dominating the left side of the view**.</think>

*Figure 12.* **Illustration of multi-hop navigation and local caption samples.** We generate 25K samples with multi-hop reasoning traces, together with 10K local captions. The local caption data focus on detailed descriptions of specific regions, such as a bounding box or half of the image.

## Data Type: Long Caption #10K

**Question: Generate an all-encompassing description of the image's visual components.**

**Answer:** The scene presents person wearing backpack and walking alongside another person.
It features a visible head, an arm, a torso, a waist, two legs, and a carried backpack, all positioned in the midground near the left side of the image. Adjacent to this figure is person wearing a blue sports shirt and blue athletic shoes. This second individual displays a head, an arm, a shirt, and shoes, and it walks in close proximity to the first person, together occupying the central depth range. To the right, person wearing a green T-shirt appears, showing only a shirt and a torso, while further behind, person showing only a face is visible with a face and a neck, and a very distant person represented by a small silhouette is seen near the bottom-left corner, contributing a faint outline without detailed parts. Near the foreground, motorcycle with a seat sits in front of the building, displaying a seat and a frame, while a second motorcycle motorcycle without visible parts is positioned slightly left of the first, both occupying the near-midground. Dominating the backdrop, building with a pinkish façade, green mosaic tiles, and a green awning stretches across the image, containing numerous components: a window with a rectangular pane on the upper left, a window with a smaller rectangular shape near the centre, and a window with a narrow vertical form on the right side; a door spanning the lower half provides an entrance; an awning covering the shop entrance shades the façade, while an second awning extending across the midsection adds depth; a signboard with decorative lettering and a second signboard with a different logo punctuate the wall; a lamp emitting soft light hangs near the awnings; a network of electrical wires criss-crossing the upper background overlays the scene; a satellite dish with a metallic surface protrudes from the roof; a massive brick wall extending from the ground to the mid-height forms the structural backdrop; and a roof with a sloping silhouette caps the building. All these elements, arranged from the immediate foreground motorcycles and walking figures through the midground persons and building components to the distant roof and brick wall, create a richly layered Moroccan market alleyway where vibrant activity, traditional architecture, and dynamic depth intertwine.

---

**Thinking:** <think>The scene presents <entity>**person wearing backpack and walking alongside another person**</entity> <bbox>[[1161,737,1317, 1147]]</bbox><depth>[[157,219]]</depth>. It features a visible head, an arm, a torso, a waist, two legs, and a carried backpack, all positioned in the midground near the left side of the image. Adjacent to this figure is <entity>**person wearing a blue sports shirt and blue athletic shoes**</entity> <bbox>[[1266,714,1397,1141]]</bbox><depth>[[126,176]]</depth>. This second individual displays a head, an arm, a shirt, and shoes, and it walks in close proximity to the first person, together occupying the central depth range. To the right, <entity>**person wearing a green T-shirt**</entity><bbox> [[1673,747,1726,1147]] </bbox><depth>[[147,171]]</depth> appears, showing only a shirt and a torso, while further behind, <entity>**person showing only a face**</entity><bbox>[[1354,818,1456,965]]</bbox><depth>[[117,135]]</depth> is visible with a face and a neck, and a very distant <entity>person represented by a small silhouette</entity><bbox>[[541,726,576,805]]</bbox><depth>[[22,24]]</depth> is seen near the bottom-left corner, contributing a faint outline without detailed parts. Near the foreground, <entity>**motorcycle with a seat**</entity><bbox>[[307,811,394,1010]] </bbox><depth>[[62,119]]</depth> sits in front of the building, displaying a seat and a frame, while a second motorcycle <entity>**motorcycle without visible parts**</entity><bbox> [[391,756,486,869]]</bbox><depth>[[35,60]]</depth> is positioned slightly left of the first, both occupying the near-midground. Dominating the backdrop, <entity>**building with a pinkish façade, green mosaic tiles, and a green awning**</entity><bbox> [[0,0,1726,1147]]</bbox><depth>[[0,254]]</depth> stretches across the image, containing numerous components: a <entity>**window with a rectangular pane**</entity><bbox>[[1132,125,1212,327]]</bbox><depth>[[80,88]]</depth> on the upper left, a <entity>**window with a smaller rectangular shape**</entity><bbox>[[901,332,943,468]]</bbox><depth>[[55,59]]</depth> near the centre, and a <entity>**window with a narrow vertical form**</entity><bbox>[[775,421,809,503]]</bbox><depth>[[44,46]] </depth> on the right side; a <entity>**door spanning the lower half**</entity><bbox>[[1606,485,1725,1087]]</bbox><depth>[[131,151]]</depth> provides an entrance; an <entity>**awning covering the shop entrance**</entity><bbox> [[1076,381,1525,699]]</bbox><depth>[[101,169]]</depth> shades the façade, while an <entity>**second awning extending across the midsection**</entity><bbox>[[610,506,894,655]]</bbox><depth>[[29,69]]</depth> adds depth; a <entity>**signboard with decorative lettering**</entity><bbox>[[495,412,579,463]]</bbox><depth>[[41,81]]</depth> and a <entity>**second signboard with a different logo**</entity><bbox> [[1133,457,1231,518]]</bbox><depth>[[78,98]]</depth> punctuate the wall; a <entity>**lamp emitting soft light**</entity> <bbox>[[1062,394,1120,531]] </bbox><depth>[[77,109]]</depth> hangs near the awnings; a <entity>**network of electrical wires criss-crossing the upper background**</entity> <bbox>[[692,0,1725,521]]</bbox><depth>[[12,166]]</depth> overlays the scene; a <entity>**satellite dish with a metallic surface**</entity><bbox> [[909,416,946,482]]</bbox><depth>[[57,61]]</depth> protrudes from the roof; a massive <entity>**brick wall extending from the ground to the mid-height**</entity><bbox>[[577,0,1725,934]]</bbox><depth>[[0,179]]</depth> forms the structural backdrop; and a <entity>**roof with a sloping silhouette**</entity><bbox>[[0,152,308,422]]</bbox><depth>[[173,254]]</depth> caps the building. **All these elements, arranged from the immediate foreground motorcycles and walking figures through the midground persons and building components to the distant roof and brick wall, create a richly layered Moroccan market alleyway where vibrant activity, traditional architecture, and dynamic depth intertwine.**</think>

*Figure 13.* **Illustration of global caption samples.** We generate 10K global captions that traverse all nodes and produce comprehensive descriptions of the entire image.

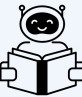

## Multi-criteria Scoring Scheme For Captioning

You are an expert evaluator for scoring captions.

Your task is to evaluate the given caption according to the criteria below. Each satisfied criterion earns 1 point.

The final output should be the total number of points satisfied.

### # Judging Criteria

- **Rule 1: Persona**: The 'think_process' must sound like a person describing an image, not a machine processing data (no "entity list", "object detected", etc.).
- **Rule 2: Formatting of Entities**: All entity references must consistently follow the '<entity>...</entity><bbox>...</bbox><depth>...</depth>' format.
- **Rule 3: Grouping of Multiple Instances**: If multiple, distinct objects of the same type exist, they must be grouped under a single descriptive block. If not applicable (only one instance of each object type exists), this rule is satisfied by default.
- **Rule 4: Tag Order & Completeness**: Every entity reference MUST strictly follow the <entity><bbox><depth> sequence. No tags should be missing within a reference.
- **Rule 5: Logical Efficiency (No Repetition)**: The same entity or object should not be described repeatedly with similar details.
- **Rule 6: Adherence to Query**: The caption must directly and accurately address the user's query.
- **Rule 7: Bounding Box Validity**: All bounding box coordinates '[x1, y1, x2, y2] (representing top-left and bottom-right corners) must be logically and numerically valid within the image dimensions. This means for 'IMAGE_DIMENSIONS of [X_max, Y_max]', the following conditions MUST be met: '0 <= x1 < x2 <= X_max and 'O <= y1 < y2 <= Y_max.
- **Rule 8: No Instance IDs**: Must not use programmatic identifiers like window_l' or 'car_2' in entity tags. NOTE: some ordinal identifiers 'a third', 'a fourth', 'a fifth' are not considered as programmatic identifiers.

### # Output Format

Just output a single number (0-8), don't output anything else.

### # Input Data:

Below is the input data. Now please make an evaluation.

**{Input Data}**

*Figure 14.* **Multi-criteria scoring scheme for captioning.** We employ a multi-criteria scoring scheme that emphasizes node coverage and edge accuracy, retaining only high-consistency caption samples.

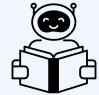

## Judge Prompt: Node-grounded Reward

You are an expert evaluator for visual analysis.

Your task is to compare the cropped image with the entity: "**{entity_name}**"

**# Judging Criteria**

**Output 1 (Match)** if ALL of the following are true:

- Entity name matches: The main subject in the image matches the entity name

- Attributes match: If the entity has attributes (color, material, etc.), they must match

- No other objects: The image contains only the target entity (blurry background is acceptable, but no other clear objects/people/animals)

**Output 0 (Not Match)** if ANY of the following is true:
- Entity name mismatch: Image shows a different object (e.g., image is "dog" but entity is "car")
- Attribute mismatch: Attributes don't match (e.g., image is "red car" but entity is "white car")
- Contains other objects: Image has other clear objects besides the target (e.g., "person + bag" when entity is "bag")

**# Examples**
✓ Image: white car → Entity: "white car" → Output: 1
✓ Image: phone → Entity: "smartphone" → Output: 1
✗ Image: dog → Entity: "car" → Output: 0 (entity name mismatch)
✗ Image: red car → Entity: "white car" → Output: 0 (attribute mismatch)
✗ Image: person holding a bag → Entity: "bag" → Output: 0 (contains other clear object)

**# Output format**
Just output a single number (1 or 0), don't output anything else.

Now please make an evaluation. Your answer:

*Figure 15.* **Judge prompt of the node-grounded reward.** More detailed examples used to instruct the MLLM are omitted for brevity.

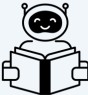

## Judge Prompt: Node-relevance Reward Prompt

You are an expert evaluator for visual relevance.

Your task is to analyze a [Question], a [Standard Answer], and a list of [Extracted Entities] to produce a precise JSON evaluation.

### # Task 1: Determine Target Count ("target_count")

- Analyze the input to determine the TOTAL number of distinct physical entities that should be grounded.
- **Rule:** Count BOTH the **Main Subject** AND any **Relational Objects/Landmarks** that **semantically support the main subject** .
- **Interaction Rule:** If the answer involves a Subject interacting with or possessing an Object (e.g., wearing, holding, containing, sitting on), **BOTH the Subject AND the Object count**.

### # Task 2: Score Each Extraction ("scores")

- Rate EACH entity in the [Extracted Entities] list against the [Standard Answer] AND [Question].
- Scoring Rubric (Strict 5-Tier System):

**1.0 — Exact / Valid / Query-Consistent**
Standard Match: Entity name and key attributes match the [Standard Answer] exactly.

**0.8 — Semantic / Attribute Similarity**
Object name or attributes are semantically similar but not identical.

**0.5 — Correct Subject, Misused Attribute**
Subject is correct, but:
- query-specific reference missing, or
- wrong attribute category used.

**0.2 — Wrong Subject**
Head noun shifts to a holder, container, or part.

**0.0 — Conflict / Negation Violation / Irrelevant**
Attribute conflict, negation violation, irrelevant object.

### # Output Format
Return ONLY a valid JSON string:
{
"target_count": <int>,
"scores": [<float>, <float>, ...]
}

### # Examples:
[Query]: What type of beverage is contained within the glass?
[Standard Answer]: The glass contains soda.
[Extracted Entities]:
- glass
- soda
Output: {"target_count": 2, "scores": [1.0, 1.0]}
...

### # Input Data:
Below is the input data. Now please make an evaluation.
**Input Data:**
{Input Data}

*Figure 16.* **Judge prompt of the node-relevance reward.** More detailed examples used to instruct the LLM are omitted for brevity.

