# Supplementary Materials of Scene Graph Thinking

In this appendix, we provide detailed implementation specifications for Scene Graph Thinking (SaGe), along with supplementary analyses and discussions that extend the main paper.

- Appendix A: Additional implementation details, including training configurations and optimization settings of SaGe.

- Appendix B: Supplementary results, including visualized QA cases and extended discussions on model generalization.

- Appendix C: Data construction pipeline and detailed composition of the large-scale structured training corpus.

- Appendix D: LLM-as-Judge prompts and reward design for node-as-proxy graph reasoning.

## A. Implementation Details

We store each scene graph in a JSON format, where every node in an image—including its category name, attributes, bounding box, depth cues, and relations to other nodes—is explicitly represented. During sampling, the stored scene graph is used as structured graph knowledge to guide our LLM-based data construction process.

We adopt Qwen2.5-VL-3B and Qwen2.5-VL-7B as our baseline models. During the cold-start stage, the models are trained for one epoch on 120K collected structured samples. The majority of the generated data are sourced from SA-1B (Kirillov et al., 2023), whose high-resolution images and rich object annotations are particularly well suited for fine-grained visual and relational analysis. The remaining 10K counting samples are collected from COCO 2014 (Lin et al., 2014). Random scaling is adopted for data augmentation. The learning rate is set to $1\mathrm{e}{-5}$ with a total batch size of 256.

In the RL post-training stage, we use a total of 26K samples, including 2K grounding samples from (Lin et al., 2014), 4K chart-related samples from (Masry et al., 2022), and 20K samples from V* following (Zheng et al., 2025). We adopt Qwen3-VL-30B-A3B-Instruct (Bai et al., 2025a) as the judge for node-grounded rewards, while Qwen3-30B-A3B-Instruct (Yang et al., 2025a) is used as the judge for both node-relevance rewards and accuracy rewards. For multiple-choice QA tasks, the accuracy reward is set to 1 if the predicted answer matches the ground truth, and 0 otherwise. For grounding-type data, we compute the IoU between the predicted bounding box and the ground-truth box, which is directly used as the reward signal. The learning rate for RL training is set to $1\mathrm{e}{-6}$ with a total batch size of 128. The model is trained for one epoch, with the number of rollouts set to 8.

## B. More Results and Discussions

### B.1. Performance Enhancement to Various Base Models

To further validate the effectiveness of the proposed Scene Graph Thinking paradigm, we conduct additional experiments on Qwen3-VL-4B and Qwen3-VL-8B. As reported in Table 7, the generated 120K structured dataset significantly improves the performance of both base models. These results demonstrate that our method generalizes well across different models and scales. It effectively enhances fine-grained visual understanding and structured reasoning capabilities.

### B.2. More Visualized QA Cases

As illustrated in Figure 7, we visualize additional QA cases to further demonstrate the effectiveness of our method. SaGe consistently exhibits stronger visual understanding and reasoning performance across a variety of tasks.

## C. Details for Constructed Data

### C.1. Data Components

To cultivate scene-graph–aligned reasoning, we develop an automated data engine to construct a large-scale training corpus consisting of 120K samples. The corpus covers diverse forms of VQA pairs and captioning data. Specifically, it includes

*Table 7.* **Performance enhancement to various base models.** With the 120K structured scene-graph data, our method consistently improves the performance of both Qwen2.5-VL and Qwen3-VL variants.

| Methods | Param Size | VstarBench | | | CVBench-2D | | | CVBench-3D | | |
|---|---|---|---|---|---|---|---|---|---|---|
| | | Overall | Attribute | Spatial | Overall | Count | Relation | Overall | Depth | Distance |
| GPT-4o (Hurst et al., 2024) | - | 67.5 | 72.2 | 60.5 | 67.5 | 72.2 | 60.5 | 65.0 | 66.8 | 63.3 |
| GPT-5-Mini (OpenAI, 2025) | - | 63.9 | - | - | - | - | - | - | - | - |
| Gemini-2.5-Pro (Google, 2025) | - | 79.2 | - | - | - | - | - | - | - | - |
| Qwen2.5-VL (Bai et al., 2025b) | 3B | 75.4 | 80.9 | 67.1 | 67.0 | 62.4 | 72.6 | 66.5 | 73.6 | 59.3 |
| Qwen2.5-VL (Bai et al., 2025b) | 7B | 76.4 | 77.4 | 75.0 | 73.6 | 62.4 | 84.9 | 73.3 | 79.8 | 66.7 |
| Qwen2.5-VL (Bai et al., 2025b) | 32B | 81.2 | 77.4 | 86.8 | 76.2 | 63.2 | 92.0 | 83.1 | 84.3 | 81.8 |
| Qwen3-VL (Bai et al., 2025a) | 4B | 80.1 | 83.5 | 75.0 | 77.1 | 65.4 | 91.4 | 92.4 | 95.0 | 89.8 |
| Qwen3-VL (Bai et al., 2025a) | 8B | 82.7 | 82.6 | 82.9 | 78.0 | 67.6 | 90.6 | 92.3 | 95.0 | 89.7 |
| **SaGe-3B (Ours)** | **3B** | 83.2 | 80.9 | 86.8 | 75.5 | 65.4 | 87.9 | 69.7 | 77.5 | 61.8 |
| **v.s. Qwen2.5-VL-3B** | **3B** | **+7.8** | **+0** | **+19.7** | **+8.5** | **+3.0** | **+15.3** | **+3.2** | **+3.9** | **+2.5** |
| **SaGe-7B (Ours)** | **7B** | 85.9 | 83.5 | 89.5 | 77.4 | 68.8 | 87.9 | 75.7 | 83.5 | 67.8 |
| **v.s. Qwen2.5-VL-7B** | **7B** | **+5.8** | **+5.6** | **+8.3** | **+7.2** | **+9.0** | **+5.5** | **+2.4** | **+3.7** | **+1.1** |
| **SaGe-4B (Ours)** | **4B** | 83.8 | 80.0 | 89.5 | 81.5 | 71.1 | 94.2 | 92.6 | 95.5 | 89.7 |
| **v.s. Qwen3-VL-4B** | **4B** | **+3.7** | **-3.5** | **+14.5** | **+4.4** | **+5.7** | **+2.8** | **+0.2** | **+0.5** | **-0.1** |
| **SaGe-8B (Ours)** | **8B** | 84.8 | 83.5 | 86.8 | 83.6 | 74.0 | 95.2 | 92.6 | 95.8 | 89.3 |
| **v.s. Qwen3-VL-8B** | **8B** | **+2.1** | **+0.9** | **+3.9** | **+5.6** | **+6.4** | **+4.6** | **+0.3** | **+0.8** | **-0.4** |