# OpenReview forum: "Scene Graph Thinking: Reinforcing Structured Visual Reasoning for Multimodal Large Language Models"
_ICML.cc/2026/Conference — ICML 2026 regular_

### Official Review · Reviewer_JXDm · 2026-02-24

**Soundness:** 3
**Presentation:** 3
**Significance:** 3
**Originality:** 4
**Overall Recommendation:** 5
**Confidence:** 4

**Summary:**

The paper presents SaGe, a new approach to empower structural reasoning for multimodal LLMs. The key novelty lies in the utilization of scene-graph representations, which store the important objects and their relations as nodes and edges, respectively. An automated generation pipeline is also introduced, in order to generate thousands of high-quality query-answer pairs, that include graph reasoning chain-of-thought, grounded on the constructed scene graphs. Finally, this generated dataset is used for supervised fine-tuning and reinforcement fine-tuning, leading to significant improvements across diverse multimodal benchmarks.

**Compliance With Llm Reviewing Policy:**

Affirmed.

**Final Justification:**

Due to the paper's original idea regarding scene graph integration in LLM reasoning and the technically sound experimental setup, my final recommendation is Accept. My concerns have been adequately addressed during rebuttal. The original score already reflects my assessment.

**Key Questions For Authors:**

Following from the weaknesses mentioned before:

1. Could the authors discuss the rationale behind limiting the scene graph edges to spatial-based relations (derived from node priors) while excluding the rich high-level semantic relations found in standard scene-graph datasets?
2. To what extent are the reported performance gains sensitive to the specific choice of these third-party models? Have any ablation studies been conducted to verify if the performance holds when these auxiliary models are swapped for equivalent alternatives, or is the gain highly dependent on this specific configuration?

**Limitations:**

The authors do not discuss the limitations of the proposed approach, such as the ones addressed previously in the weaknesses section. The paper would highly benefit from a Limitation section, addressing the weaknesses mentioned previously, as well as strict limitations/requirements regarding the hardware (such as GPU/VRAM), since the proposed methodology includes several training stages and inference of numerous different models. This would also be useful information for anyone who would like to reproduce the experiments of the paper.

**Strengths And Weaknesses:**

Strengths:
1. The main idea presented in the paper is original and opens up an interesting field of research for empowering MLLM reasoning by using graph-based information. Both the main motivation and the intuition behind the proposed approach are presented clearly and coherently. All the diagrams also present helpful examples, and assist the reader in understanding this complex pipeline.
2. The experiments are conclusive of the importance of the proposed method. A wide variety of both open-source models (Qwen, InternVL, LLava, etc.) as well as proprietary MLLMs (GPT-4o, GPT-5-Mini, Gemini-2.5-Pro), are sufficient to showcase the impact of SaGe. Similarly, a wide variety of benchmarks (focused on fine-grained perception, spatial reasoning and general visual comprehension) is utilized for this specific task, further boosting the authors' claim.
3. The ablation study particularly shows that most of the proposed additions and ideas by the authors have non-trivial impact on the final outcome
4. All sufficient details for reproducible results are mentioned in the paper and the appendices, including specific models used, training schemes and prompts.

Weaknesses:
1. The main weakness of the paper is the under-utilization of the semantic capabilities that scene graphs can offer. Specifically, the majority of the most widely-used scene graph datasets [1,2,3] include edges that go beyond simple spatial-based relations between objects (i.e. "on", "below", "in front of", "to the right of"), which are the only ones that can be extracted from the node-level priors that are extracted from each node (Line 189: bounding boxes, depth cues, and component structure). Traditional scene graphs also contain plenty of high-level relations between objects (e.g. "person playing-with person", "person swinging baseball-bat", "person riding horse"). Such high-level relations are found across all the most common scene-graph datasets, and usually constitute the majority of relation classes. Since a big part of the evaluation consists of general comprehension tasks, it seems that this limitation will create a bottleneck for the model’s reasoning abilities. Additionally there is a variety of SotA models that specialize on Scene Graph Generation [3], and could automatically provide you with high-level relations, at no extra training cost.
2. (Minor) There seems to be excessively high dependence on other LLMs in the entire pipeline (several models for generation, for general prompting, for extracting similarity, for LLM-as-a-Judge, for the rewards, for dataset validation). This seems to be an important factor in the proposed pipeline, and while the final results are certainly showing a significant performance gain, it raises a concern, as to how much of this increase in evaluation metrics would be diminished (or amplified) simply by changing all these third-party models that the current pipeline relies on.
3. (Minor) The indirect utilization of the scene-graphs should be explicitly stated at the beginning of Section 3, or in the Introduction. Throughout the introduction, the method is described as 'reasoning via explicit scene-graph representations,' which implies direct input. However, Figures 2 and 3 reveal that the scene graphs are not actually included in the MLLM context. This discrepancy may cause confusion until the reader reaches the final examples in Section 3.


Additional Notes:
- The appendices in the supplementary material, should be added to the main paper, as there is unlimited space for appendices in the template, below the references

[1] Krishna, Ranjay et al. “Visual Genome: Connecting Language and Vision Using Crowdsourced Dense Image Annotations.” International Journal of Computer Vision 123 (2016): 32 - 73.

[2] Hudson, Drew A. and Christopher D. Manning. “GQA: A New Dataset for Real-World Visual Reasoning and Compositional Question Answering.” 2019 IEEE/CVF Conference on Computer Vision and Pattern Recognition (CVPR) (2019): 6693-6702.

[3] Yang, Jingkang et al. “Panoptic Scene Graph Generation.” ArXiv abs/2207.11247 (2022): n. pag.

---

> ### Author Rebuttal · Authors · 2026-03-31
>
> Thank you for your insightful advice and for acknowledging our contribution and novelty. We will revise it accordingly.
>
> >**Under-utilization of Scene Graph**
>
> Thank you for this insightful and helpful comment. In our edge-centric query sampling, we capture not only spatial layout relations but also eight higher-level semantic relations (e.g., holding, wearing, touching, action influence, using, sharing, connecting, and ownership). These relations help MLMMs efficiently navigate to the target nodes. We are sorry for this confusion and will clarify it further in the main text.
>
> We appreciate the suggestion to leverage existing scene graph generation models, which is indeed helpful for constructing primary-level entities. However, our scene graphs focus on fine-grained secondary sub-entities, enabling MLLMs to achieve more fine-grained perception and complex visual reasoning, which is a level of granularity where existing SOTA models are less effective. In future work, we plan to integrate both approaches to further improve the efficiency and quality of our graph construction pipeline.
>
> >**Sensitivity to Different External Models**
>
> Thank you for raising this important concern. Although using LLMs for data generation and as a judge is a common practice to reduce human cost, our main contribution lies in introducing a hierarchical, graph-based approach to mine structured relations, enabling fine-grained and structured reasoning. For the data construction, more advanced models can generate higher-quality data and ensure reliable performance. Considering the cost-effectiveness and reliability, we use open-source models for scene graph construction, with Gemini employed only for filtering.
>
> For LLM-as-judge, we alternate between Qwen2.5-72B-Instruct and Qwen3-30B-A3B. As shown in Tab. S6, performance across VStarBench, CV-2D, and CV-3D remains largely consistent, indicating that our **gains stem from the hierarchical graph-based design rather than the specific choice of judge model**. In general, as long as the judge provides reasonably reliable reward signals, i.e., assessing whether the model outputs are semantically consistent with the ground truth, the impact on model training tends to be minimal.
>
> - Tab. S6. Sensitivity to Different External Models
>
> | Judges        |   V*  | CV-2D | CV-3D |
> |---------------|:-----:|:-----:|:-----:|
> | Qwen2.5-72B   | 89.5  | 77.9  | 71.8  |
> | Qwen3-30B-A3B | 89.0  | 77.8  | 72.3  |
>
> >**Method Clarity**
>
> Thank you for pointing out this clarity issue. We have revised the Introduction and the beginning of Section 3 to explicitly state that the scene graph serves as a structured prior for generating graph-aligned training data. It guides the construction of reasoning trajectories, enabling the model to perceive objects in context, reason over relationships, and traverse structured paths for fine-grained visual reasoning.
>
> >**Appendix Placement**
>
> Thank you for pointing out this issue. We have moved the appendix to follow the main paper after the references, improving clarity and readability.
>
> >**Limitation Discussion**
>
> Thank you for your helpful suggestion. We will add a Limitations section to discuss potential constraints of our approach. In particular, our method involves multiple models and training stages, which require relatively high computational resources. In future work, we plan to develop an agentic, end-to-end scene graph paradigm to reduce construction costs. In addition, we will provide more detailed implementation information, including hardware specifications, to further improve reproducibility.

---

> > ### Author Rebuttal · Reviewer_JXDm · 2026-04-01
> >
> > Thank you to the authors for their detailed rebuttal. My concerns have been adequately addressed, but the score already reflects my assessment, so I will not adjust it.

---

> > > ### Author Response · Authors · 2026-04-03
> > >
> > > Thank you for your positive feedback and for taking the time to review our rebuttal.
> > > We are pleased that our responses have addressed your concerns.
> > > We will carefully incorporate the suggested clarifications and improvements in the final version to further strengthen the paper.

---

### Official Review · Reviewer_Y6C6 · 2026-03-10

**Soundness:** 3
**Presentation:** 3
**Significance:** 3
**Originality:** 3
**Overall Recommendation:** 5
**Confidence:** 5

**Summary:**

Conventional "thinking-with-images" methods only focus on *isolated objects* and neglect structured relationships. Therefore, this paper introduces Scene Graph Thinking (SaGe), where the authors first represent each training sample as a scene-graph and build 120K high-quality training samples, and next propose two node-related rewards for GRPO. Empirically, SaGe models surpass base models by a large margin, and achieves competitive results with conventional "think-with-images" methods.

**Compliance With Llm Reviewing Policy:**

Affirmed.

**Final Justification:**

Please check the "Rebuttal Acknowledgement" section.

**Key Questions For Authors:**

The advantages compared with other "think-with-images" methods seem to be limited (Table 1). This might be because of insufficient evaluations. It is encouraged to evaluate on more advanced fine-grained perception benchmarks. Moreover, qualitative comparisons and detailed analysis of SaGe and conventional "think-with-images" methods are also missing.

**Limitations:**

No.

**Strengths And Weaknesses:**

**Strengths**
1. This paper is well-written and easy to follow
2. The motivation is clear and reasonable.
3. The conducted data seem to be of high quality.
4. Improvements over baselines are significant.


**Weaknesses**
1. The advantages compared with other "think-with-images" methods seem to be limited (Table 1). This might be because of insufficient evaluations. It is encouraged to evaluate on more advanced fine-grained perception benchmarks, e.g., MME-RealWorld [R1], TreeBench [R2], and VisualProbe [R3]. Moreover, qualitative comparisons of SaGe and conventional "think-with-images" methods are also missing.
2. Potential reward hacking: as most rewards are computed by MLLMs. Is there any risk of reward hacking, and how can this be alleviated?
3. How to conduct text ground-truth of SFT data seems missing.

**References**

[R1] Mme-realworld: Could your multimodal llm challenge high-resolution real-world scenarios that are difficult for humans? ICLR, 2025.

[R2] Traceable evidence enhanced visual grounded reasoning: Evaluation and methodology. ICLR, 2025.

[R3] Mini-o3: Scaling up reasoning patterns and interaction turns for visual search. arXiv, 2025.

---

> ### Author Rebuttal · Authors · 2026-03-31
>
> Thank you for your insightful advice and for acknowledging our contribution. We will revise it accordingly.
>
> >**Comparisons with Think-with-Images Methods on More Advanced Benchmarks**
>
> Thank you for this helpful suggestion. We evaluated our method on more complex real-world benchmarks, including MME-RealWorld, TreeBench, and VisualProbe. Our approach achieves **competitive performance** compared to conventional think-with-image methods, as reported in Tab. S5.
>
> Moreover, as suggested by Reviewer aGW1, SaGe demonstrates **improved performance-efficiency trade-offs over DeepEyes** with significantly less inference time and consumed tokens. (reported in Tab. S2).
>
> - Tab. S5. Comparisons with Think-with-Images Methods on More Advanced Benchmarks.
>
> | Methods | Param Size | | MME-RealWorld_EN || TreeBench || VisualProbe ||
> |--------------------|:---------------:|:----------------:|:----------:|:---------:|:---------:|:-----------:|:------:|:-----:|
> |||Overall | Perception | Reasoning |Overall| Hard| Medium |Easy |
> | gpt-4o|-| 62.8 |64.9|47.3|46.9|11.2 |15.4| 47.5|
> | LLaVA-OV|7B | 57.4 |- | - |37.3|13.4 |12.5| 36.2|
> | LLaVA-OV| 72B | 63.2 |- | - |40.5|19.8 |21.6| 51.8|
> | InternVL3|8B | 62.0 |- | - |38.8|10.4 |12.7| 42.6|
> | InternVL3| 78B | 65.4 |- | - |46.4|17.0 |15.7| 43.3|
> | Qwen2.5-VL|3B | 55.8 |58.1|38.7|36.0|24.9 |30.2| 43.2|
> | Qwen2.5-VL|7B | 57.4 |59.6|41.0|37.0|31.1 |30.2| 44.0|
> | Qwen2.5-VL| 32B | 60.5 |68.0|44.4|43.2|33.0 |34.0| 44.0|
> | Pixel-Reasoner |7B | 64.4 |- | - |39.0|28.8 |29.6| 58.4|
> | DeepEyes|7B | 64.0 |66.6|44.7|37.5|35.1 |29.8| 60.1|
> | Thyme|7B | 64.8 |67.1|48.4|37.8|33.0 |30.6| 50.4|
> | **SaGe_3B** |**3B**|**60.0**|**62.2**|**44.2**|**36.5**|**33.1**|**34.0**|**59.6**|
> | v.s. Qwen2.5-VL-3B |3B | +4.2 |+4.1|+5.5|+0.5| +8.2|+3.8| +16.4 |
> | **SaGe_7B**|**7B**|**65.4**|**67.8**|**47.8**|**37.8**|**35.9**|**38.1**|**63.8**|
> | v.s. Qwen2.5-VL-7B |7B | +8.0 |+8.2|+6.8|+0.8| +4.8|+7.9| +19.8 |
>
> >**Qualitative Comparisons with Think-with-Images Method and Detailed Analysis**
>
> Qualitative comparisons between SaGe and the think-with-image method DeepEyes, are shown in Fig. S3 and Fig. S4. While DeepEyes can handle fine-grained perception, we observe limitations in its CoT reasoning.
>
> **First, limited perceptual ability**. In Fig. S3, although DeepEyes attends to the correct region (broom), it still mispredicts the color, leading to an incorrect outcome.
>
> **Second, redundant tool usage**. In Fig. S4, even after correctly identifying the target region in the first tool call, an additional unnecessary call is made. These issues limit DeepEyes fine-grained performance and introduce significantly higher inference overhead.
>
>  **Third, substantial inference cost**. Beyond the high token consumption from cropping and instructions, a large amount of redundant and uninformative text, along with repeated tool tags embedded in the CoT (filtered in the figures for better readability), further increases token overhead.
>
> In contrast, our method leverages structured graph information to enhance fine-grained perception and reasoning in MLLMs. The SFT stage internalizes graph-aligned reasoning, while in the RL stage, the node-grounded reward encourages accurate perception of visual attributes and the node relevance reward reduces unnecessary traversal of irrelevant nodes. These designs improve fine-grained understanding and lower inference cost, offering a promising direction to address the limitations of think-with-image methods.
>
> - Fig. S3 and S4. Qualitative Comparisons with Think-with-Images Method DeepEyes.
>
> [Fig. S3](https://anonymous.4open.science/api/repo/ICML_Response-5979R/file/Fig_S3.pdf?v=ce80f1c9)
>
> [Fig. S4](https://anonymous.4open.science/api/repo/ICML_Response-5979R/file/Fig_S4.pdf?v=8e40428c)
>
> >**Potential Reward Hack**
>
> Thank you for raising this concern. We observed reward hacking during reward design. When applying the node-grounded reward alone, optimizing for alignment between node text and visual attention caused the model to favor obvious, simple but irrelevant nodes, leading to ineffective navigation. To address this, we introduce a node relevance reward, which preserves flexible graph reasoning while preventing spurious rewards and guiding the model more effectively toward the target.
>
> >**Text Ground-Truth for SFT Data**
>
> Thank you for this helpful comment. Given the generated scene graphs, we automatically construct large-scale SFT data via LLM-guided graph sampling to foster hierarchical reasoning. The queries, CoT steps, and text ground-truth are all obtained through carefully designed LLM prompting. To ensure data quality, we also apply a two-stage verification process with human-cross validation, which effectively filters noisy samples and guarantees reliable training data. We will clarify it more clearly in the revised manuscript.

---

> > ### Author Rebuttal · Reviewer_Y6C6 · 2026-04-04
> >
> > All my concerns have been addressed. I am extremely excited to see that the proposed method achieves competitive performance on recent advanced fine-grained visual reasoning benchmarks and performs much more efficiently. Therefore, I will raise my rating from 4 to 5.
> >
> > The authors are expected to revise the paper based on this discussion.

---

> > > ### Author Response · Authors · 2026-04-04
> > >
> > > Thank you for your positive feedback and for updating your rating. We are glad that the effectiveness and efficiency of our method are recognized. We will carefully revise the paper based on this discussion to further improve the clarity and quality of the manuscript.

---

### Official Review · Reviewer_qAJU · 2026-03-12

**Soundness:** 3
**Presentation:** 3
**Significance:** 2
**Originality:** 2
**Overall Recommendation:** 4
**Confidence:** 4

**Summary:**

This paper proposes Scene Graph Thinking (SaGe), a post-training framework for MLLMs that aims to improve fine-grained and structured visual reasoning by converting flat image-text data into hierarchical scene graphs. The pipeline includes: (1) automatic scene-graph construction; (2) generation of 120K graph-aligned training samples with node-articulated CoT; (3) a two-stage training recipe consisting of SFT followed by GRPO-based RL with two node-level rewards. Experiments on high-resolution, spatial, and general visual benchmarks show substantial gains over Qwen2.5-VL baselines.

**Compliance With Llm Reviewing Policy:**

Affirmed.

**Final Justification:**

The author's rebuttal solved most of my concern.

**Key Questions For Authors:**

1. Can the authors provide a direct quantitative evaluation of scene-graph quality, including node precision/recall, relation accuracy, and error breakdowns for sub-entities, depth cues, and edges?

2. How much of the final gain comes from the scene-graph representation itself versus simply adding richer CoT supervision with entity/bbox/depth tags? A stronger controlled comparison against non-graph structured annotations would help.

**Limitations:**

Yes

**Strengths And Weaknesses:**

Strengths
------------
1. Clear problem setting.
The paper targets an important weakness of current “think-with-image” MLLMs: they often rely on local crops and heuristic search while underusing scene structure and inter-object relations.

2. Technically coherent framework.
The full pipeline from graph construction to graph-sampled data generation to graph-aligned RL is conceptually consistent, and the node-as-proxy reward is a reasonable way to avoid brittle edge-level supervision.

3. Useful ablations.
The paper separates the effects of cold-start SFT, GRPO, CoT structure. The ablations support that the gains are not from a single component alone.

Weaknesses
------------
1. The method depends heavily on strong proprietary or external components.
Scene graph construction and verification rely on Qwen2.5-VL-72B, GPT-OSS-120B, a proprietary MLLM judge, Depth Anything, and SAM. This makes the claimed contribution partly a system/data-engineering result, and weakens reproducibility.

2. Core graph quality is insufficiently validated. The paper argues that the graph is the key inductive structure, yet there is little direct evaluation of graph correctness, edge quality, or failure modes beyond a brief statement of less than 1% failure rate under random human cross-validation.

3. Potential fairness and data-overlap concerns are not fully clarified. The RL stage uses 20K samples from the VStarBench training set, while VStarBench is also a major reported evaluation benchmark. The paper likely uses train/test separation, but this should be stated much more clearly.

4. More benchmarks should be included. Other reasoning-based benchmarks like MathVista, MMMU should be included to evaluate the generality of the proposed method.

---

> ### Author Rebuttal · Authors · 2026-03-31
>
> Thank you for your constructive suggestions and for acknowledging our contribution. We will revise it accordingly.
>
> >**Dependence on External Components**
>
> The graph construction involves carefully designed processing. However, when the curated graph data is generated, it significantly empowers the baseline models with fine-grained understanding and visual reasoning, which shows a better performance–efficiency trade-off than think-with-image methods (shown in Tab. S2). As noted by Reviewer JXDm, this suggests a promising direction for enhancing MLLM reasoning via graph-based information.
>
> To make it more user-friendly, we will release all constructed data and code to facilitate result replication.
>
> >**Scene Graph Quality**
>
> Thank you for this constructive suggestion. To assess scene graph quality, we evaluate **1,000 sampled raw graphs** using Gemini-3-Pro, measuring the correctness of **22029 nodes** and **26283 edges**. A node is considered correct if its bounding box region matches its entity tag and attributes. It reports that the nodes and edges in our constructed graph exhibit high precision, **with 94.5% of nodes and 92.9% of edges correctly identified**.
>
> Regarding node recall and depth, exhaustive node annotation is prohibitively challenging due to the open-ended and hierarchical nature of scene graphs. Our depth signals provide range cues for depth reasoning rather than precise estimation, and their effectiveness is supported by strong performance on the depth-related benchmark CVBench-3D (Table 2 in the main text).
>
> We also provide error breakdowns in Fig. S1 and Fig. S2, where **52.0% of node errors** arise from mismatches between node attributes and visual regions, and **55.6% of edge errors** stem from invalid relations. In our data construction pipeline, we employ a two-round verification process, consisting of a GME-based filtering step followed by MLLM re-verification during QA construction. Based on the generated high-quality scene graphs and this verification process, we obtain curated data with a failure rate of less than 1% under random human cross-validation.
>
> - Fig. S1 and S2. Quantitative Evaluation of Scene Graph
>
> [Fig. S1](https://anonymous.4open.science/api/repo/ICML_Response-5979R/file/Fig_S1.pdf?v=32e526a1)
>
> [Fig. S2](https://anonymous.4open.science/api/repo/ICML_Response-5979R/file/Fig_S2.pdf?v=4b88e0a5)
>
>
> >**Gain from Scene Graph**
>
> Thank you for this helpful suggestion. To isolate the effect of supervision with entity/bbox/depth tags, we conduct a controlled experiment using only node-centric queries on primary-level entities, excluding edge-based queries such as multi-hop reasoning, interactions, or captions, resulting in 120K samples. Tab. S3 shows that our full method, which incorporates both node- and edge-centric reasoning via the scene graph, outperforms this ablation, demonstrating that the scene-graph representation itself contributes substantially beyond simply adding richer CoT supervision.
>
> - Tab. S3. Gains from Scene Graph. 3B is used.
>
> ||V*| CV-2D | CV-3D |
> |----------|:----:|:-----:|:-----:|
> | No Graph |80.6|73.2|67.1|
> |**Ours**|**83.2**|**75.5**|**69.7**|
>
> >**More Benchmarks**
>
> Thank you for this helpful suggestion. Unlike methods such as DeepEyes and Thyme that incorporate math-centric data during training, our scene graphs are designed for vision-centric scenes. Our graph construction and curated data do not involve math tasks, which may lead to a performance drop on MathVista, as shown in Tab. S4. Nevertheless, on diverse real-world visual benchmarks, including MMU, MME-RealWorld, TreeBench, and VisualProbe (as suggested by Reviewer Y6C6), our method achieves strong results, demonstrating its effectiveness and generality for fine-grained visual perception and understanding.
>
> - Tab. S4. Performance on More Benchmarks.
>
> | Methods| Size | MathVista |MMMU | MME-Real | TreeBench | VisualProbe_Hard | VisualProbe_Medium | VisualProbe_Easy |
> |--------------------|:----:|:---------:|:-----:|:--------:|:---------:|:-----:|:-----:|:---:|
> | gpt-4o|-|63.4|69.1 |62.8|46.9|11.2 |15.4 |47.5 |
> | Qwen2.5-VL|3B|59.3|44.3 |55.8|36.0|24.9 |30.2 |43.2 |
> | Qwen2.5-VL|7B|68.1|49.2 |57.4|37.0|31.1 |30.2 |44.0 |
> | Qwen2.5-VL|32B|75.5|58.1 |60.5|43.2|33.0 |34.0 |44.0 |
> | Pixel-Reasoner|7B|-|-|64.4|39.0|28.8 |29.6 |58.4 |
> | DeepEyes|7B|69.3|50.5 |64.0|37.5|35.1 |29.8 |60.1 |
> | Thyme |7B|70.0|-|64.8|37.8|33.0 |30.6 |50.4 |
> | **SaGe_3B** |**3B**|**57.7**|**45.2**|**60.0**|**36.5**|**33.1**|**34.0**|**59.6**|
> | v.s. Qwen2.5-VL-3B |3B|-1.6|+0.9|+4.2|+0.5|+8.2 |+3.8 |+16.4|
> | **SaGe_7B**|**7B**|**66.8**|**50.3**|**65.4**|**37.8**|**35.9**|**38.1**|**63.8**|
> | v.s. Qwen2.5-VL-7B |7B|-1.3|+1.1|+8.0|+0.8|+4.8 |+7.9 |+19.8|
>
> >**Data Overlap**
>
> Thank you for raising this concern. VStarBench has strict official train/test splits. Our 20K RL samples are drawn only from the training set, ensuring no overlap with evaluation data. This separation will be clearly stated in the revised manuscript.

---

> > ### Author Rebuttal · Reviewer_qAJU · 2026-04-03
> >
> > Thanks for the rebuttal, which solves most of my concern.

---

> > > ### Author Response · Authors · 2026-04-03
> > >
> > > Thank you for your positive feedback and for taking the time to review our rebuttal.
> > > We are glad that our responses have addressed most of your concerns, and we will further improve the manuscript accordingly. We would be grateful if you could take these clarifications into account in your final assessment.

---

### Official Review · Reviewer_aGW1 · 2026-03-13

**Soundness:** 3
**Presentation:** 2
**Significance:** 3
**Originality:** 3
**Overall Recommendation:** 5
**Confidence:** 4

**Summary:**

This paper proposes a new paradigm for training MLLMs for visual reasoning using structured CoT patterns obtained from a scene graph. To this end, the paper proposes a pipeline for automatically collecting scene graph data obtained from image-text pairs and augmenting it with additional object parts, depth information, and object relationships. The paper then generates questions related to the nodes and edges in the graphs, including multi-hop questions, and performs data validation. Finally, the paper proposes a two-stage training method consisting of cold-start data containing graph-based CoT patterns, which is augmented by GRPO-fine-tuning using rewards that encourage traversing relevant graph nodes during reasoning.

**Compliance With Llm Reviewing Policy:**

Affirmed.

**Final Justification:**

The rebuttal presented by the authors addressed my comments on the paper. Overall, the paper is well-written, and the method is sufficiently original and effective. As a result, my confidence in my evaluation has increased.

**Key Questions For Authors:**

- Explicit Reasoning increases inference time. It would be good to measure the quality of question answers to understand tradeoffs between QA quality and inference time. It would also be helpful to understand how much resources are required to reproduce the paper's results and whether a public release of the dataset and/or code is planned.
- Since generating additional tokens increases model inference time, is it possible to dynamically determine the quIn my opinion, the author should have added a mention of the computational overhead that Structural Reasoning adds over the basic model.estion type so that the model understands whether it needs to use a certain part of the graph data or not?

**Limitations:**

In my opinion, the author should have added a mention of the computational overhead that Structural Reasoning adds over the basic model.

**Strengths And Weaknesses:**

Strengths:
- The paper demonstrates a strong empirical result, namely, that structured reasoning significantly improves the response quality of baseline models.
- A large number of ablations were performed, and the improvement in reasoning quality is noticeable across various datasets.
- Despite the expensive training dataset, SaGe does not require additional labeling during inference and works directly with images.

Weaknesses:
- The method description only becomes clear towards the end. Especially the graph's role remains unclear until the part where it is clear that the graph is essentially used to generate data for additional training (questions and CoT for Cold Start). The description of the method and the figures should be made clearer.
- Table 4 shows that the effects of adding entity, bbox, and depth blocks are somewhat specific to the validation dataset.

---

> ### Author Rebuttal · Authors · 2026-03-31
>
> Thank you for your insightful advice and for acknowledging our novelty and contribution. We will revise it accordingly.
>
> >**Method Clarity**
>
> Thank you for pointing out this clarity issue. We have revised the method description to highlight the central role of the scene graph from the beginning. Specifically, our scene-graph representation serves as a structured data prior for generating reasoning trajectories, enabling the model to perceive objects in context, reason over relationships, and traverse structured pathways for more reliable visual reasoning. We have also improved the organization and updated the figures accordingly in the Introduction and Method parts.
>
> >**Generalization Across Tasks**
>
> Table 4 is designed to explicitly disentangle the roles of different CoT elements. These patterns provide grounded evidence for graph-based reasoning, enabling strong generalization beyond the listed validation sets to broader visual scenarios, such as chart understanding (ChartQA) and general visual reasoning (MMStar).
>
> Moreover, as suggested by Reviewer Y6C6, we validate SaGe across diverse real-world benchmarks, such as MME-RealWorld, TreeBench, and VisualProbe. The consistent performance in Tab. S1 further validates the generality of our scene graph-based representation.
>
> - Tab. S1. Performance Across Diverse Benchmarks.
>
> | Methods | Param Size | | MME-RealWorld_EN || TreeBench || VisualProbe ||
> |--------------------|:---------------:|:----------------:|:----------:|:---------:|:---------:|:-----------:|:------:|:-----:|
> |||Overall | Perception | Reasoning |Overall| Hard| Medium |Easy |
> | gpt-4o|-| 62.8 |64.9|47.3|46.9|11.2 |15.4| 47.5|
> | LLaVA-OV|7B | 57.4 |- | - |37.3|13.4 |12.5| 36.2|
> | LLaVA-OV| 72B | 63.2 |- | - |40.5|19.8 |21.6| 51.8|
> | InternVL3|8B | 62.0 |- | - |38.8|10.4 |12.7| 42.6|
> | InternVL3| 78B | 65.4 |- | - |46.4|17.0 |15.7| 43.3|
> | Qwen2.5-VL|3B | 55.8 |58.1|38.7|36.0|24.9 |30.2| 43.2|
> | Qwen2.5-VL|7B | 57.4 |59.6|41.0|37.0|31.1 |30.2| 44.0|
> | Qwen2.5-VL| 32B | 60.5 |68.0|44.4|43.2|33.0 |34.0| 44.0|
> | Pixel-Reasoner |7B | 64.4 |- | - |39.0|28.8 |29.6| 58.4|
> | DeepEyes|7B | 64.0 |66.6|44.7|37.5|35.1 |29.8| 60.1|
> | Thyme|7B | 64.8 |67.1|48.4|37.8|33.0 |30.6| 50.4|
> | **SaGe_3B** |**3B**|**60.0**|**62.2**|**44.2**|**36.5**|**33.1**|**34.0**|**59.6**|
> | v.s. Qwen2.5-VL-3B |3B | +4.2 |+4.1|+5.5|+0.5| +8.2|+3.8| +16.4 |
> | **SaGe_7B**|**7B**|**65.4**|**67.8**|**47.8**|**37.8**|**35.9**|**38.1**|**63.8**|
> | v.s. Qwen2.5-VL-7B |7B | +8.0 |+8.2|+6.8|+0.8| +4.8|+7.9| +19.8 |
>
> >**Dynamic Reasoning and Computational Overhead**
>
> Thank you for this helpful suggestion. To avoid unnecessary multi-hop reasoning, the constructed CoT does not involve graph traversal for simple first-level entities. The multi-hop reasoning traces are generated only for challenging entities, such as small-scale sub-entities. This design enables the model to dynamically perform graph traversal for challenging entities while maintaining efficient inference.
>
> Tab. S2 reports that our model improves fine-grained understanding with only marginal overhead in token usage and inference time. Compared to the think-with-image baseline DeepEyes, our method achieves competitive performance with significantly lower inference cost.
>
> - Tab. S2. Inference Efficiency on VStarBench. All methods are deployed with vLLM on a single GPU, without concurrent inference.
>
> | Methods       | Inference Time (min) | Token Usage | Accuracy  |
> |---------------|:--------------------:|:-----------:|:---------:|
> | Qwen2.5-VL-3B |         3.5          |     382     |   75.4    |
> | Qwen2.5-VL-7B |         4.0          |     382     |   76.4    |
> | DeepEyes-7B   |         52.7         |   5631538   |   85.6    |
> | **SaGe_3B**       |         **4.0**          |    **20188**    |   **89.0**    |
> | **SaGe_7B**       |         **6.3**          |    **21226**    |   **89.0**    |
>
>
> >**Required Resources**
>
> Thank you for this helpful suggestion. For the GPU resources, 8 GPUs are used for the 7B training during the SFT stage. 8 GPUs are used for policy optimization and 16 GPUs for the Judge deployment in the RL stage. This setup is relatively more accessible compared to traditional think-with-image methods, such as DeepEyes. All the data sources are collected from public datasets.
>
> To facilitate reproducibility, we will release all constructed data and code upon acceptance of the paper.

---

> > ### Author Rebuttal · Reviewer_aGW1 · 2026-04-03
> >
> > Thanks to the authors for the detailed answer, it answered my concerns.

---

> > > ### Author Response · Authors · 2026-04-04
> > >
> > > Thank you for your positive feedback and for taking the time to review our rebuttal.
> > > We are pleased that our responses have addressed your concerns.
> > > We will incorporate the suggested clarifications in the final version to further improve the paper.

---

### Decision · Program_Chairs · 2026-04-30

**Decision:**

Accept (regular)

**Comment:**

Overall, the paper proposes a coherent and timely approach to improving structured visual reasoning in MLLMs via scene-graph-based data construction and training. All reviewers agree that the problem is important and that the method leads to consistent empirical gains across a range of benchmarks. The rebuttal effectively addressed most concerns, and all reviewers maintained or increased their confidence, leading to a generally positive consensus.

In terms of strengths, the work presents a well-motivated framework that integrates scene graph construction, graph-aligned data generation, and a two-stage training strategy in a conceptually consistent way. Empirical results are quite good, with clear improvements over competitive baselines and ablation studies supporting the contribution of each component. The idea of leveraging structured graph representations to guide reasoning is considered useful for multimodal reasoning.

However, several weaknesses remain. First, clarity and presentation were initially an issue, particularly regarding the role of the scene graph (as a data-generation prior rather than a direct input), though this is largely addressed in the rebuttal. Second, the approach depends heavily on external models and a complex data pipeline, which raises concerns about reproducibility and how much of the gain is attributable to engineering choices versus core methodological innovation. Third, evaluation could be more comprehensive: some reviewers noted limited comparisons on certain reasoning benchmarks, insufficient qualitative analysis, and a lack of deeper validation of scene graph quality and failure modes. Additional concerns include potential data overlap risks (clarified in rebuttal), limited exploration of richer semantic relations, and lack of detailed discussion on computational overhead and resource requirements.

Overall, despite these limitations, the paper provides a solid and well-validated contribution with clear empirical benefits and a promising direction for structured reasoning in MLLMs. The concerns are mostly incremental and can be addressed in the final version.